# D-serine reconstitutes synaptic and intrinsic inhibitory control of pyramidal neurons in a neurodevelopmental mouse model for schizophrenia

Xiao-Qin Zhang [1], Le Xu [1], Xin-Yi Zhu [1], Zi-Hang Tang[1], Yi-Bei Dong[1], Zhi-Peng Yu[1], Qing Shang [2], Zheng-Chun Wang[1] & Hao-Wei Shen [1] ✉

The hypothesis of N-methyl-D-aspartate receptor (NMDAR) dysfunction for cognitive impairment in schizophrenia constitutes the theoretical basis for the translational application of NMDAR co-agonist D-serine or its analogs. However, the cellular mechanism underlying the therapeutic effect of D-serine remains unclear. In this study, we utilize a mouse neurodevelopmental model for schizophrenia that mimics prenatal pathogenesis and exhibits hypoexcitability of parvalbumin-positive (PV) neurons, as well as PV-preferential NMDAR dysfunction. We find that D-serine restores excitation/inhibition balance by reconstituting both synaptic and intrinsic inhibitory control of cingulate pyramidal neurons through facilitating PV excitability and activating small-conductance $Ca^{2+}$-activated $K^+$ (SK) channels in pyramidal neurons, respectively. Either amplifying inhibitory drive via directly strengthening PV neuron activity or inhibiting pyramidal excitability via activating SK channels is sufficient to improve cognitive function in this model. These findings unveil a dual mechanism for how D-serine improves cognitive function in this model.

Abnormalities of parvalbumin-positive (PV) neurons in the frontal cortex represent a prominent pathology in schizophrenia patients[1] and animal models for schizophrenia[2]. These abnormalities include reductions in glutamate decarboxylase, $GABA_A$ receptor and GABA membrane transporter in PV neurons, as well as decreased PV density and mRNA expression[3]. Especially, multiple lines of evidence using transgenic mouse models support the hypothesis that hypofunction of the N-methyl-D-aspartate receptor (NMDAR) in cortical PV neurons is linked to schizophrenia pathogenesis[4,5]. Such molecular alterations in PV neurons disturb the local excitation/inhibition (E/I) balance and neural oscillation, and impair inhibitory control of cortico-cortical communication and cortical-subcortical output[6,7], which is associated with the pathophysiological mechanism for cognitive impairment in schizophrenia[8].

Clinical trials have shown that stimulating the NMDAR 'glycine modulatory site' using D-serine, glycine, or glycine transporter 1 (GlyT1) inhibitor to enhance NMDAR activity has significant beneficial effects on cognitive function in schizophrenia[9–11], further supporting the hypothesis of NMDAR hypofunction in schizophrenia. In the framework of hypothesis of NMDAR deficit in PV neurons, D-serine or its analogs should be apt to activate NMDAR in PV neurons in order to play a therapeutic role. However, studies have revealed that NMDAR has little effect on synaptic activation of PV neurons[12–14] due to its sparse expression in PV neurons[15] and slow kinetics[16], and rapid chelation of $Ca^{2+}$ influx by PV[17,18]. Therefore, the ability of D-serine to regulate PV function via activating NMDAR remains uncertain. Moreover, the extensive distribution of NMDAR and the complicated intracellular signaling triggered by NMDAR-mediated $Ca^{2+}$ influx imply

[1]Department of Pharmacology, School of Medicine, Ningbo University, 818 Fenghua Rd, Ningbo, Zhejiang 315211, China. [2]Department of Neurology, The First Affiliated Hospital of Ningbo Univerisity, 59 Liuting Street, Haishu District, Ningbo, Zhejiang 315211, China. ✉e-mail: shenhaowei@nbu.edu.cn

that the therapeutic effect of D-serine should involve multiple pathways. Thus, a long-standing question is how D-serine or its analogs restore the disrupted E/I balance in the local circuitry by precisely modulating inhibitory control mechanisms. Clarifying these questions help us understand the pathophysiological basis of cognitive dysfunction in schizophrenia and the essential mechanisms underlying the therapeutic effect of D-serine, thereby facilitating the discovery of new pharmacotherapeutic targets.

Pathological alterations in the anterior cingulate cortex (ACC) are associated with impairment in social cognition and negative symptoms of schizophrenia[19,20]. Neuroimaging and postmortem studies have revealed various abnormalities in the ACC of schizophrenia patients, including reduced volumes, abnormal activities[21], decreased mRNA expression of NMDAR in interneurons[22]. In the rodent models for schizophrenia, studies have reported a reduction in the number of PV neurons in the ACC, along with decreased perineuronal net surrounding PV neurons and increased oxidative stress level in PV neurons[23].

Methylazoxymethanol acetate (MAM) is a DNA-alkylating agent that interferes with DNA synthesis in proliferating cells[24]. To model neurodevelopmental disorders, pregnant rodents are administered with MAM during a key gestational window, disrupting offspring neurodevelopment[25,26]. In this study, MAM model served as a neurodevelopmental model for schizophrenia. MAM model suffered from hypoexcitability of PV neurons with NMDAR hypofunction, resulting in reduced inhibitory output to pyramidal neurons in the ACC. Importantly, we discovered that D-serine reconstitutes synaptic and intrinsic inhibitory control of pyramidal neurons via modulating the intrinsic properties of PV neurons and activating specific K⁺ channels in pyramidal neurons, respectively. These findings reveal the cellular mechanism through which D-serine improves cognitive function and suggest a novel pharmacological target for the treatment of schizophrenia model.

## Results

### D-serine inhibited the intrinsic excitability of pyramidal neurons without affecting PV neurons in the control group

We first investigated how D-serine differentially regulated the excitability of pyramidal and PV neurons in the ACC of the control group, which referred male and female offspring of pregnant dams treated with saline (Fig. 1a). As shown in Fig. 1b, c, D-serine exhibited a significant inhibitory effect on the intrinsic excitability of cingulate pyramidal neurons (two-way ANOVA with repeated measures; detailed statistics shown in the corresponding legend). Interestingly, we observed that D-serine enhances the amplitude of medium-duration afterhyperpolarization (mAHP) that followed a single action potential (Fig. 1f). Generally, mAHP is largely modulated by small-conductance $Ca^{2+}$-activated K⁺ (SK) channels and limits firing frequency[27]. Apamin diminished the effects of D-serine on mAHP, firing frequency, rheobase and resting membrane potential (RMP) (one-way ANOVA with repeated measures, Fig. 1g–i), verifying that SK channel mediated the inhibitory effect of D-serine on pyramidal neurons. Additional membrane properties regulated by D-serine are shown in Supplementary Fig. 1a–f. Furthermore, we excluded a role for large-conductance $Ca^{2+}$-activated K⁺ channels (BK) channels in D-serine-modulated pyramidal excitability (Supplementary Fig. 1g–p).

In cingulate PV neurons, D-serine did not have an overall impact on the intrinsic excitability of PV neurons (Fig. 1e; two-way ANOVA with repeated measures), while it caused an increase in input resistance and a decrease in rheobase (paired t-test, Fig. 1j–m). D-APV did not alter firing frequency of PV neurons under baseline conditions or during D-serine incubation, suggesting that NMDAR is unlikely to contribute to these processes (Fig. 1n, Two-way ANOVA).

Due to the fast-spiking properties, mAHPs were unmeasurable in PV neurons in present study. Indeed, the parameters of membrane properties we observed in PV neurons did not provide sufficient clues to identify specific ion channels that may be involved in the effect of D-serine on PV excitability.

The E/I balance in the local cortical circuit determines the homeostasis of cortico-cortical communication and the efficiency of cortical-subcortical output. Thus, we further observed the effect of D-serine on evoked action potential (eAP) that integrated excitatory and inhibitory synaptic inputs to form an output signal, representing the local E/I balance. Figure 1o shows eAP elicited by a train of 10 stimuli at 20 Hz in the ACC of the control group. The current intensity of stimulation was adjusted for each neuron to induce 4−6 spikes. D-serine did not affect eAP frequency for each stimulus (post hoc test with two-way ANOVA, Fig. 1p) or the average eAP number of each neuron (post hoc test with one-way ANOVA, Fig. 1q), while blocking inhibitory input with picrotoxin significantly enhanced eAP responses.

### Chronic D-serine treatment ameliorated cognitive dysfunction in a neurodevelopmental mouse model for schizophrenia

Gestational MAM administration is utilized to model prenatal neurodevelopmental disorders, providing an animal model that reflects the neurodevelopmental hypothesis of schizophrenia. To evaluate the cognitive function of these mice, sensorimotor gating, spatial working memory and recognition memory that were examined by PPI, Y-maze and NLR/NOR tests, respectively. To improve the cognitive function of the MAM group, D-serine was added into drinking water as a chronic treatment for at least 2 weeks, resulting in a daily dose of about 100 mg/kg D-serine, which has been proven to increase D-serine levels in the brain[28]. Animals were divided into four groups, including control and MAM groups, each with or without D-serine treatment (Fig. 2a). Consistent with previous findings[29,30], the MAM group exhibited normal levels of acoustic startle responses (Fig. 2b), but decreased PPI (Fig. 2c). The MAM group showed no abnormality in locomotor activity (Fig. 2d). In Y-maze tests, the total number of individual arm entries in the MAM group was comparable to controls (Fig. 2e), but the percentage of spontaneous alternations was decreased in the MAM group (Fig. 2f). In both NLR and NOR tests, a decreased discrimination index was found in MAM group (Fig. 2g, h). In the aforementioned behavioral tests, oral supplementation of D-serine in the MAM group had significant therapeutic effects on sensorimotor gating, spatial working memory and recognition memory, but had no effects on startle responses and locomotor activity (interaction of prenatal exposure × treatment by two-way ANOVA, Fig. 2b–h).

### MAM group suffered from PV hypofunction and diminished inhibitory control in the ACC microcircuit

Next, we investigated the neuropathophysiological alterations in the cingulate neurons of the MAM group. The MAM group exhibited a significant reduction in intrinsic excitability in PV (two-way ANOVA with repeated measures, Fig. 3a, b), but not in pyramidal neurons (Supplementary Fig. 2a, b). Additionally, the rheobase was increased while RMP and input resistance were decreased in ACC PV neurons of the MAM group (unpaired t-test, Fig. 3c–f). We further evaluated the NMDAR function of PV and pyramidal neuron via analyzing NMDAR-mediated sEPSC charge[31]. Compared to control group, MAM group showed less NMDAR-mediated sEPSC charge (Fig. 3i), and higher ratio of the AMPAR- to the NMDAR-mediated sEPSCs (A/N ratio) (Fig. 3j) in PV but not in pyramidal neurons (Supplementary Fig. 2c−e, unpaired t-test). These results indicate that MAM group suffered from hypoexcitability of PV neuron with NMDAR hypofunction, while its pyramidal neurons retained relatively intact intrinsic and synaptic properties. This neurodevelopmental model supports the hypothesis of PV neuron dysfunction for schizophrenia.

To examine the effect of PV neuron dysfunction on inhibitory output to pyramidal neurons, ChR2 was selectively expressed in the ACC of MAM PV-Cre mice via a local injection of AAV2/9-hEF1a-DIO-

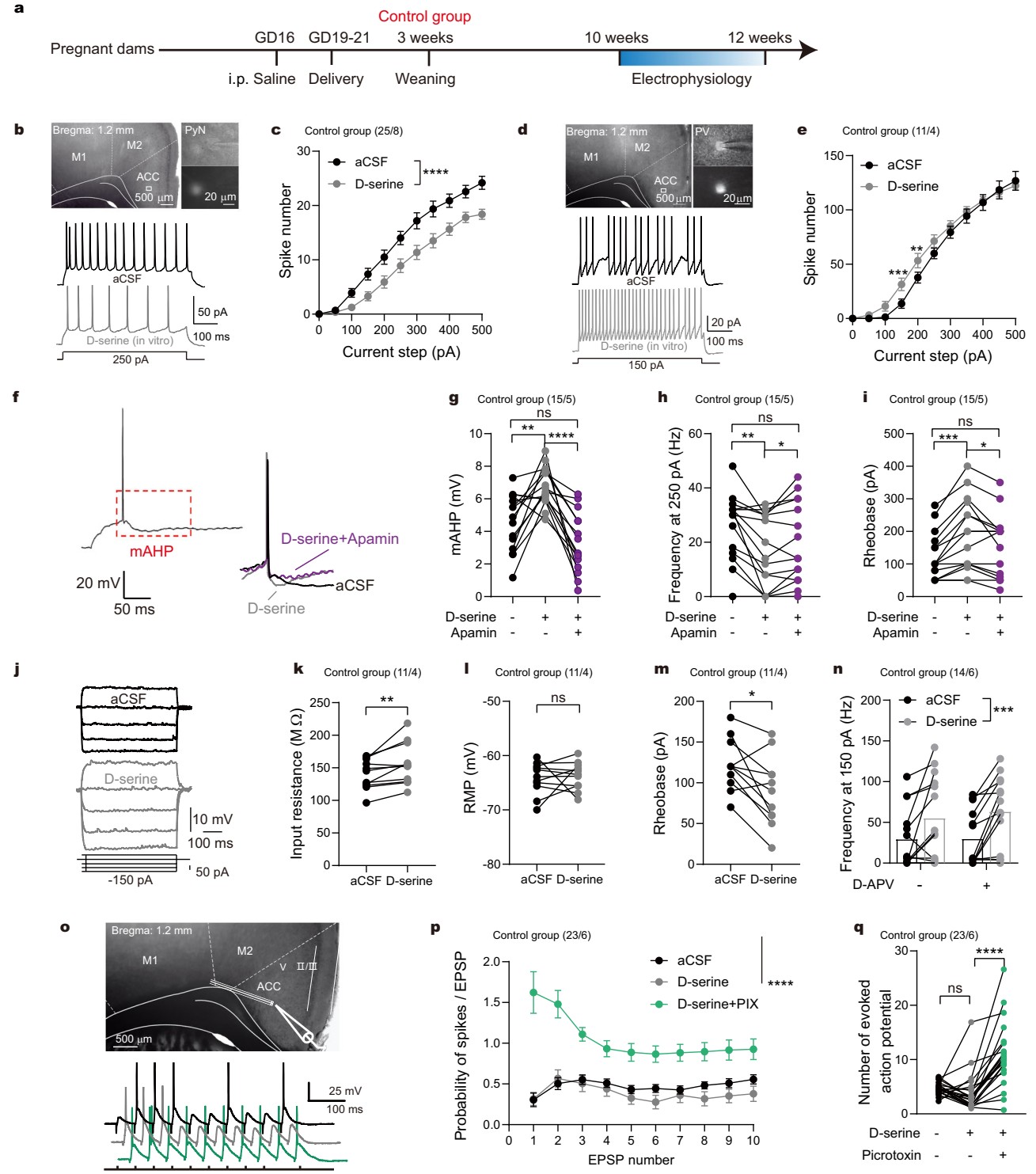

ChR2(H134R)-mCherry (Supplementary Fig. 3). The amplitudes of inhibitory postsynaptic currents optogenetically (oIPSCs) evoked by four 5-ms pulses of blue laser with 10 mW at 20 Hz were measured from pyramidal neurons (Fig. 4a, b). Compared to the control group, the MAM group exhibited a lower release response to the 1st stimulus and less release inhibition to succeeding stimuli (unpaired *t*-test or *post hoc* test with two-way ANOVA, Fig. 4c–e). The higher ratio of oIPSC amplitude evoked by the subsequent stimulus vs. the 1st stimulus can reflect a lower initial probability of neurotransmitter release from PV neurons. Moreover, a significant decrease in the frequency of sIPSCs to pyramidal neurons adjacent to mCherry-

labeled PV neurons was recorded in the MAM group, while no effect was observed in the mean amplitude of sIPSCs (unpaired *t*-test for average, Fig. 4f–h). These results suggest that PV neuron dysfunction results in attenuated inhibitory output, which might weaken inhibitory control over pyramidal neurons and disturb the E/I balance in the local microcircuit. To validate this possibility, we further investigated the synapse-driven pyramidal activity (i.e. eAP). Pyramidal neurons in the MAM group exhibited a significantly increased probability of burst firing compared to the control group (chi-square test, Fig. 4i, j), indicating impaired E/I balance and disinhibition in the local ACC circuit of the MAM group.

**Fig. 1 | D-serine modulated intrinsic excitability distinctively in cingulate PV and pyramidal neurons. a** Schematic of the control experiment setup. **b, d** Brain slice images displaying ACC and neuron recording sites (square) with overlaid infrared-DIC and fluorescence; typical action potential traces pre-/post-D-serine. **c, e** D-serine significantly decreased spike numbers in pyramidal neurons (two-way ANOVA: drug $F_{(1, 24)} = 35.17$), without no overall effect on PV neurons (drug $F_{(1, 10)} = 2.30$), as shown in the summarized current-firing curves. ****$p < 0.0001$ indicating the impact of D-serine (**c**); **$p = 0.0014$, ***$p = 0.0002$, Bonferroni's *post hoc* test (**e**). **f** Representative action potential from a pyramidal neuron with analyzed mAHP (red dotted line). **g–i** D-serine modulation of mAHP, firing frequency, and rheobase in pyramidal neurons was prevented by apamin (**$p = 0.0041$, ****$p < 0.0001$; *$p = 0.0421$, **$p = 0.0032$; *$p = 0.0291$, ***$p = 0.0006$; Bonferroni's tests). **j** Voltage responses to depolarizing current steps in PV neurons. **k–m**

D-serine increased input resistance (two-sided paired *t* test: $t_{(10)} = 3.224$, **$p = 0.0091$) and decreased rheobase ($t_{(10)} = 3.080$, *$p = 0.0116$) in PV neurons. **n** D-APV did not affect firing frequency of PV neurons to 150 ρA depolarizing current (D-APV $F_{(1, 26)} = 0.084$, $p = 0.774$; D-serine $F_{(1, 26)} = 20.45$, $p = 0.0001$). **o, p** No change in eAP frequency with D-serine and increases with PIX, shown in preparation schematics and eAP traces (interaction of drug × stimulus $F_{(2.698, 56.35)} = 7.055$, $p = 0.0006$; drug $F_{(1.831, 40.29)} = 32.29$, $p < 0.0001$; stimulus $F_{(1.935, 42.58)} = 5.595$, $p = 0.0075$). **q** Mean eAP numbers remain unchanged with D-serine (****$p < 0.0001$, Bonferroni's test). Data are shown as mean ± SEM in (**c, e, n**). The first and second N numbers in parentheses correspond to the numbers of recorded neurons and mice, respectively. Source data are provided as a Source Data file. GD gestational day, ACC anterior cingulate cortex, M1/M2 primary/secondary motor cortex, PyN pyramidal neuron, PV parvalbumin, mAHP medium afterhyperpolarization, PIX picrotoxin.

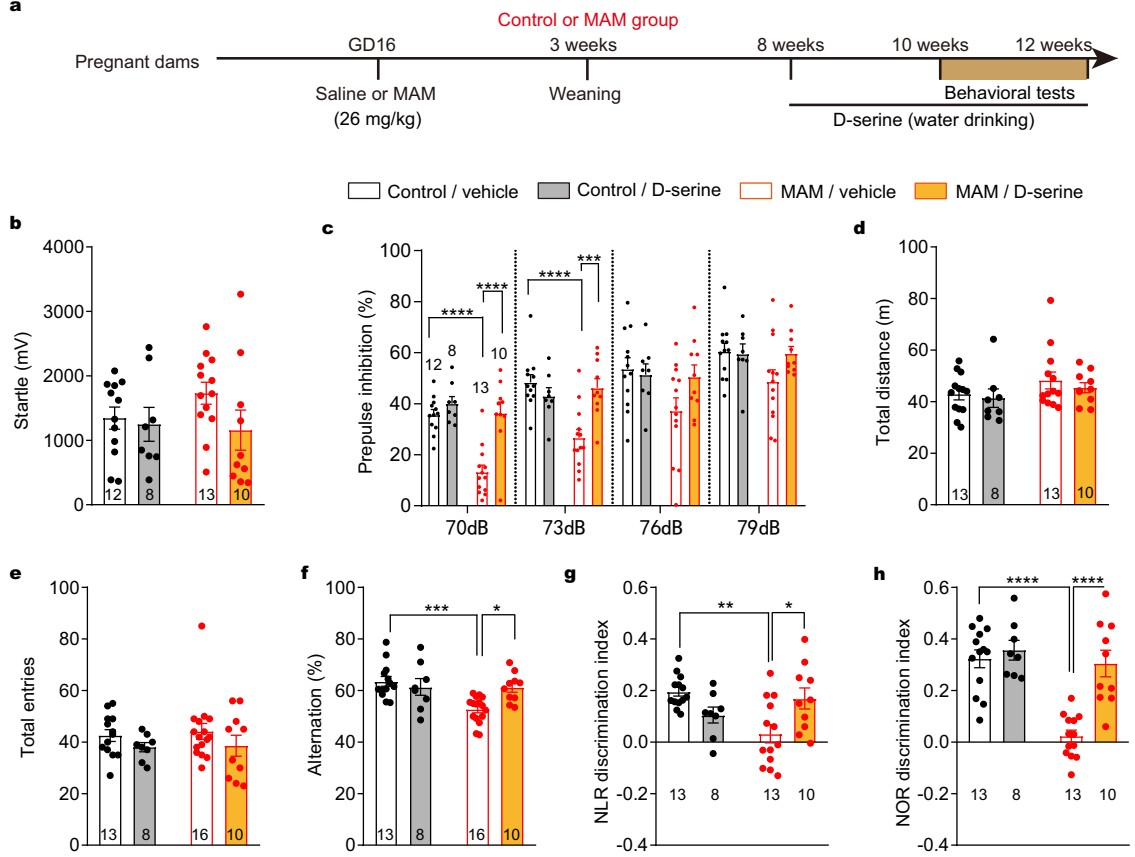

**Fig. 2 | D-serine ameliorated cognitive deficits in MAM group. a** Schematic of experiment setup for MAM group. **b** Neither prenatal MAM nor D-serine affected startle response (two-way ANOVA: MAM × D-serine $F_{(1, 39)} = 1.097$, $p = 0.301$, MAM $F_{(1, 39)} = 0.424$, $p = 0.519$, D-serine $F_{(1, 39)} = 2.163$, $p = 0.149$). **c** D-serine restored PPI deficits in MAM group (70 dB: MAM x D-serine $F_{(1, 39)} = 9.242$, $p = 0.004$, MAM $F_{(1, 39)} = 18.28$, $p = 0.0001$, D-serine $F_{(1, 39)} = 19.97$, $p < 0.0001$; 73 dB: MAM x D-serine $F_{(1, 39)} = 13.59$, $p = 0.0007$, MAM $F_{(1, 39)} = 7.475$, $p = 0.009$, D-serine $F_{(1, 39)} = 4.350$, $p = 0.044$). **d** Neither prenatal MAM nor D-serine affected locomotor activity (MAM x D-serine $F_{(1, 40)} = 0.053$, $p = 0.819$, MAM $F_{(1, 40)} = 2.681$, $p = 0.109$, D-serine $F_{(1, 40)} = 0.578$, $p = 0.452$;). **e** Neither prenatal MAM nor chronic D-serine affected total entries in Y-maze tests (MAM x D-serine $F_{(1, 43)} = 0.024$, $p = 0.878$, MAM $F_{(1, 43)} = 0.118$, $p = 0.733$, D-serine $F_{(1, 43)} = 2.498$, $p = 0.121$). **f** D-serine improved spatial working memory in MAM group (MAM x D-serine $F_{(1, 43)} = 7.323$, $p = 0.010$, MAM $F_{(1, 43)} = 8.140$, $p = 0.007$, D-serine $F_{(1, 43)} = 0.549$, $p = 0.118$). **g, h** D-serine improved recognition memory in MAM group (NLR test: MAM x D-serine $F_{(1, 40)} = 12.17$, $p = 0.001$, MAM $F_{(1, 40)} = 2.310$, $p = 0.136$, D-serine $F_{(1, 40)} = 0.491$, $p = 0.487$; NOR test: MAM x D-serine $F_{(1, 40)} = 10.98$, $p = 0.002$, MAM $F_{(1, 40)} = 21.98$, $p < 0.0001$, D-serine $F_{(1, 40)} = 17.63$, $p = 0.010$). *$p < 0.05$, **$p < 0.01$, ***$p < 0.001$, ****$p < 0.0001$ Bonferroni's *post hoc* test. Data are shown as mean ± SEM. N numbers shown in or under bars correspond to the number of mice assigned in behavioral experiments. Source data are provided as a Source Data file. NLR Novel Location Recognition; NOR Novel Object Recognition.

## D-serine strengthened synaptic and intrinsic inhibitory control of cingulate pyramidal neurons in the MAM group

Because the MAM group exhibited NMDAR hypofunction and hypoexcitability in PV neurons, we asked whether NMDAR co-agonist D-serine could restore PV function in the MAM group. Bath application of D-serine significantly promoted the PV excitability in a NMDAR-dependent manner in the MAM group (Fig. 5a–c, two-way and one-way ANOVA). Additionally, the rheobase was decreased

while input resistance was increased after D-serine application (Supplementary Fig. 4a–d). We further compared the effect of D-serine on intrinsic excitability between the control and MAM groups. Two-way ANOVA analysis revealed that D-serine has an overall impact on spike number elicited by 250 ρA of depolarizing current, irrespective of prenatal exposure (Fig. 5d). Importantly, depolarizing current-induced spikes in the MAM group were more sensitive to D-serine than in the control group (*post hoc* test, Fig. 5d).

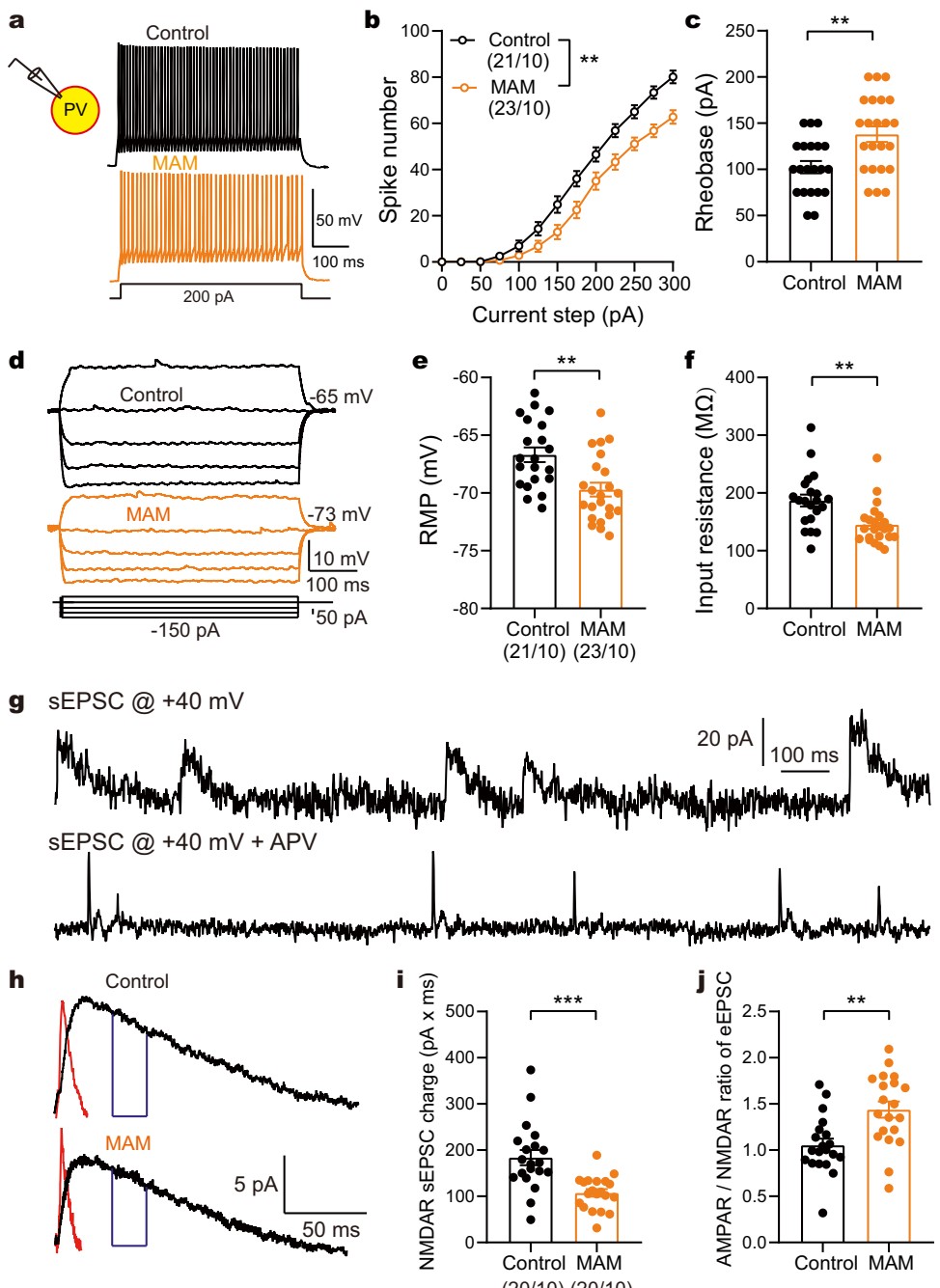

**Fig. 3 | MAM group exhibited hypoexcitability and NMDAR deficit in PV neurons in the ACC. a** Representative traces of action potential elicited by a depolarizing current step in PV neurons. **b**, **c** A reduced spike number and an increased rheobase in PV neurons were observed in MAM group. **p = 0.0019 indicating the impact of prenatal MAM on spike numbers of PV neurons (two-way ANOVA: $F_{(1, 42)} = 11.00$); **p = 0.0021 (two-sided unpaired $t$ test: $t_{(42)} = 3.274$). **d** Representative traces of voltage responses to a depolarizing from −150 to 50 ρA current step in the above PV neurons. **e**, **f** An increased RMP (two-sided unpaired $t$ test: $t_{(42)} = 3.458$, **p = 0.0013) and a decreased input resistance ($t_{(42)} = 3.322$, **p = 0.0019) of PV neurons were observed in MAM group. **g** Raw traces of dual sEPSCs recorded at +40 mV and pure

AMPAR sEPSCs isolated after bath application of the NMDAR antagonist D-APV (100 μM). **h** Typical averaged events of NMDAR and AMPAR-mediated sEPSC in PV neuron in the ACC. **i**, **j** MAM group showed lower NMDAR sEPSCs charge (two-sided unpaired $t$ test: $t_{(38)} = 4.176$, ***p = 0.0002) and higher AMPAR/NMDAR ratio of sEPSC ($t_{(38)} = 3.455$, **p = 0.0014) in MAM group than in control group. Data are shown as mean ± SEM. The first and second N numbers in parentheses correspond to the numbers of recorded neurons and mice, respectively. Source data are provided as a Source Data file. RMP resting membrane potential, sEPSC spontaneous excitatory postsynaptic current, AMPAR α-amino-3-hydroxy-5-methyl-4-isoxazolepropionic acid receptor, NMDAR, N-methyl-D-aspartate receptor.

And D-serine brought spike frequency at 250 ρA of depolarizing current in the MAM group back to the same level as in the control group (*post hoc* test, Fig. 5d). Furthermore, D-serine significantly enhanced NMDAR-mediated sEPSC charge and decrease the A/N ratio in the MAM group (interaction of prenatal exposure × drug application by two-way ANOVA, Fig. 5, f). These results suggest that

hypoexcitability of PV neurons might be attributable to NMDAR deficiency, which could be compensated by D-serine.

In cingulate pyramidal neurons, similar to the results observed in the control group, D-serine application exhibited apamin-sensitive effects on the intrinsic excitability and other membrane properties in the MAM group (Fig. 5g–j), indicating a SK channel-mediated

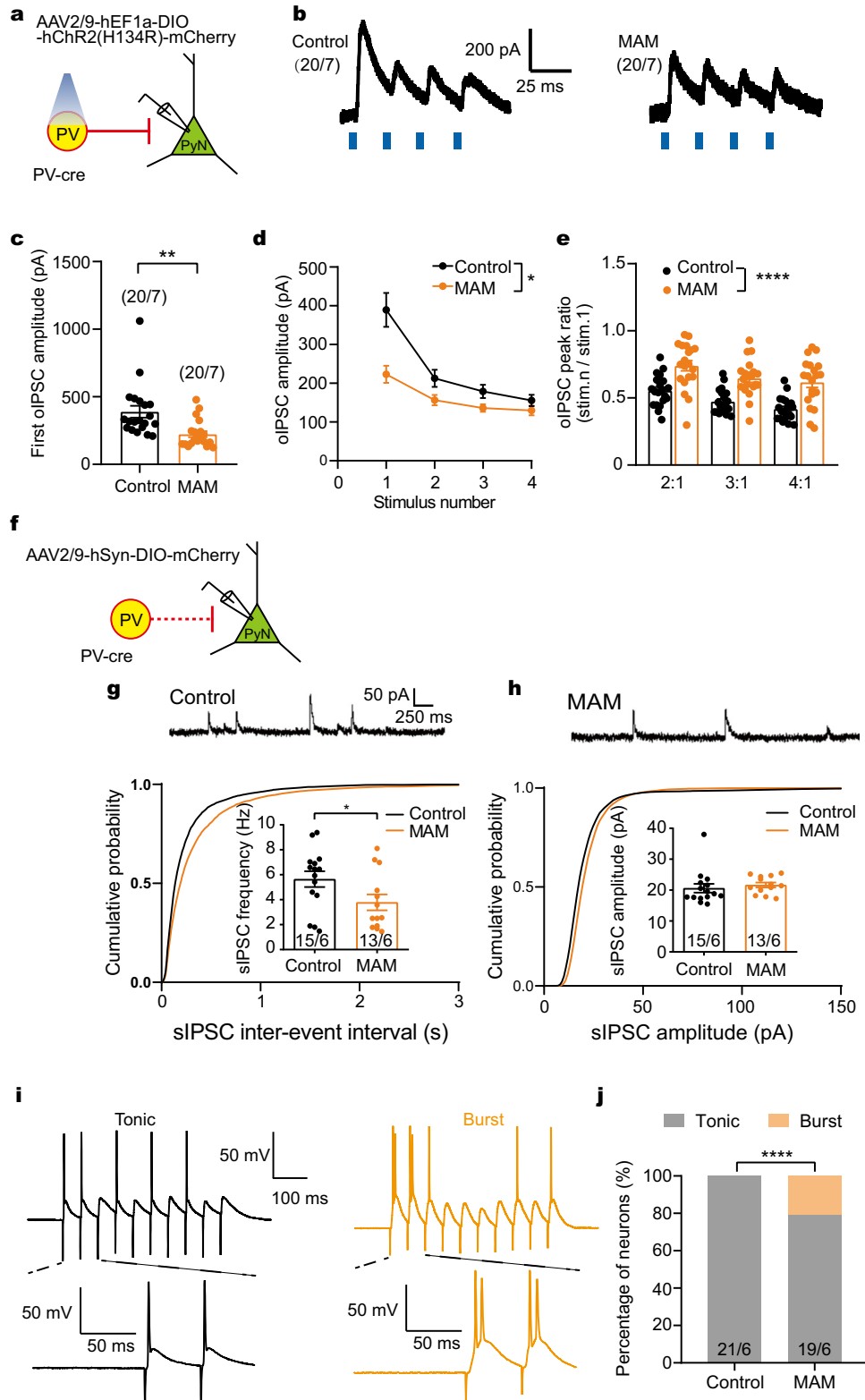

regulation. Additionally, D-serine exhibited an overall impact on spike number elicited by 250 pA of depolarizing current, irrespective of prenatal exposure (Fig. 5k, Two-way ANOVA).

Given the abilities of D-serine in regulating the excitability of both PV and pyramidal neurons in the MAM group, we then tested whether D-serine improves the impaired the E/I balance in the ACC of the MAM group. The average number of eAPs was significantly reduced (*post hoc* test with one-way ANOVA, Fig. 5l), and no inducible burst firing during

stimulus trains was observed after D-serine incubation in the MAM group. These results suggest that D-serine strengthens inhibitory control from PV neurons and suppresses synapse-driven pyramidal activity in the ACC of the MAM group.

We further investigated the effects of systemic D-serine adminis-tration on PV excitability and synapse-driven pyramidal activity in the MAM group. These mice were treated with D-serine in their drinking water for a minimum of 2 weeks (Fig. 6a). The *post hoc* test following

**Fig. 4 | Pyramidal neurons received less inhibitory input from PV neurons in MAM group. a** Diagram of AAV expressing Cre-dependent ChR2 stereotactically injected to the ACC of PV-cre mice. PV neurons were specifically activated by blue laser at 473 nm and oIPSC was recorded from pyramidal neurons. **b** Representative traces of PV neurons-derived oIPSCs evoked by blue laser at 20 Hz in control or MAM groups. **c, d** The amplitude of oIPSC evoked by the 1st stimulus in MAM group was less that in control group (two-sided unpaired $t$ test: $t_{(38)} = 3.374$, **$p = 0.0017$). And MAM group displayed less release inhibition to succeeding stimuli (Two-way ANOVA: MAM x stimulus $F_{(3, 114)} = 13.46$, $p < 0.0001$, MAM $F_{(1, 38)} = 7.13$, $p = 0.011$, stimulus $F_{(1.243, 47.24)} = 73.92$, $p < 0.0001$), compared to control group. **e** MAM group showed higher ratio of oIPSC amplitude evoked by the subsequent stimulus vs. the 1st stimulus (MAM $F_{(1, 38)} = 23.46$, $p < 0.0001$, stimulus $F_{(1.871, 71.1)} = 38.48$,

$p < 0.0001$). **f** Pyramidal neurons adjacent to the mcherry-labeled PV neurons were used to record spontaneous postsynaptic current. **g, h** Representative traces of sIPSCs are shown on the top. MAM group exhibited less sIPSC frequency (g, two-sided unpaired $t$ test: $t_{(26)} = 2.059$, *$p = 0.0496$) and unchanged amplitude (h, $t_{(26)} = 0.6185$), compared to control group. **i** Different patterns of tonic (left) and burst (right) firing during eAP recording in pyramidal neurons. **j** Percentage of neurons exhibiting either tonic or burst firing in all recorded neurons in control or MAM group (two-sided Chi-square test, ****$p < 0.0001$). Data are shown as mean ± SEM except **j**. The first and second N numbers in parentheses or bars correspond to the numbers of recorded neurons and mice, respectively. Source data are provided as a Source Data file. oIPSC optically evoked inhibitory postsynaptic current.

one-way ANOVA showed that the firing frequency in the MAM group treated with D-serine was statistically comparable to that in the control group, implying a rebound effect. There was no significant difference between the D-serine-treated and water-treated MAM groups, suggesting that systemic D-serine partially restored the PV excitability in the MAM group to the level observed in the control group (Fig. 6d). Moreover, only one out of 19 sampled neurons exhibited burst firing induced by the eAP protocol in the MAM group receiving systemic D-serine, suggesting a lower probability of burst firing compared to the MAM group receiving only daily water (chi-square test, Fig. 6e, f). These findings suggest that systemic D-serine administration reestablishes the disrupted E/I balance in the local ACC circuit of the MAM group.

**Chemogenetic activation of PV neurons ameliorated cognitive dysfunction in the MAM group**
The above results demonstrate that PV dysfunction is the core pathophysiology in the ACC of the MAM group, and that strengthening PV inhibitory control to pyramidal neurons with D-serine could be an effective way to treat the impaired cognition. However, damage to other types of GABAergic interneurons cannot be neglected in the MAM group, and D-serine might improve cognitive performance via benefiting the functional recovery of those neurons. This raised a critical question: is rescuing PV neurons sufficient to ameliorate cognitive dysfunction in the MAM group? Therefore, we selectively repaired PV hypoexcitability in the MAM group through a chemogenetic strategy. Enhanced excitability and decreased rheobase of PV neurons with CNO application confirmed that Gq-coupled hM3Dq DREADD functionally expressed in PV neurons (Fig. 7a–f). All behavioral tests were conducted 30 min after CNO (2 mg/kg, i.p.) or saline (10 mL/kg, i.p.) treatment (Fig. 7g). The DREADD-expressing MAM group exhibited improvements in sensorimotor gating, spatial working memory and recognition memory (interaction of DREADD × CNO by two-way ANOVA, Fig. 7h–n). These results suggest that rescuing PV dysfunction is sufficient to restore the impaired cognition of the MAM group. Supplementary Fig. 5 reveals that CNO does not affect PPI and cognitive performance in the hM3D-expressing control group, indicating that the behavioral improvements observed in the MAM model are due to the specific correction of PV hypofunction.

**SK channel activator riluzole improved the impaired cognitive function of the MAM group**
D-serine apparently suppressed the excitability of pyramidal neurons potentially mediated by SK channel and improved the impaired cognitive function of the MAM group, indicating the involvement of SK channel in therapeutic effect. The clinically available drug riluzole is a SK channel activator[32]. Figure 8a–f shows that riluzole, in an apamin-sensitive manner, enhanced mAHP and inhibited pyramidal excitability in the ACC (one-way ANOVA with repeated measures), confirming riluzole's action in cingulate pyramidal neuron through the modulation of SK channel. Therefore, we then examined whether riluzole plays a similar role to D-serine in cognitive improvement. Riluzole

significantly improved sensorimotor gating, spatial working memory and recognition memory in the MAM group but not in the control group (interaction of prenatal exposure × treatment by two-way ANOVA, Fig. 8g–n), implying that SK channel could be a potential pharmacotherapeutic target for impaired cognitive performance in schizophrenia.

## Discussion
The NMDAR 'glycine modulation site' is a direct or indirect common target for D-serine, D-cycloserine, glycine transport protein 1 inhibitors and D-amino acid oxidase inhibitors, which are promising agents for improving cognitive impairment in schizophrenia[33,34]. In this study, we utilized a neurodevelopmental model for schizophrenia to investigate the cellular mechanism by which D-serine improves cognitive function in schizophrenia. The pathophysiological alterations in the ACC of MAM group were manifested in the hypoexcitability, abnormal membrane properties and reduced NMDAR-mediated currents in PV neurons of the ACC, and resulting the attenuation in inhibitory output to pyramidal neurons. These findings suggest a diminished inhibitory control in cingulate microcircuit in the MAM model, and support the hypothesis of PV neuron hypofunction for schizophrenia. This study is the first to reveal that, relative to pyramidal neurons, NMDAR deficits preferentially occur in PV neurons in a non-genetic mouse model for schizophrenia mimicking prenatal pathogenesis. Intriguingly, our study discovered that D-serine could facilitate inhibitory control in the ACC of MAM group via a dual mechanism. The ability of D-serine to promote PV excitability was strengthened in the MAM group, which promoted inhibitory output to pyramidal neuron. And D-serine directly suppressed pyramidal excitability potentially mediated by SK channels. Consequently, D-serine reinstituted synaptic and intrinsic inhibitory control of cingulate pyramidal neurons and shifted E/I balance in the ACC of MAM group. These results denote that the D-serine acts on both PV and pyramidal neurons through multiple targets to ameliorate cognitive impairment in neurodevelopmental disorders in which NMDAR deficit-induced PV neuron dysfunction is a key pathology. A key consideration here is the heterogeneity amongst the deep layer pyramidal neurons within the prefrontal cortex concerning their projection targets and physiological properties[35]. This diverse population may present differential effects within a neurodevelopmental model of schizophrenia and varying responses to D-serine interventions. Our approach to randomly sample these neurons was to ensure a holistic view of the pyramidal neuron population in the ACC across the control and MAM groups.

Our finding suggests that NMDAR dysfunction in PV neuron is critical factor contributing to cognitive impairment in the MAM model. However, previous studies have demonstrated that genetic deletion of NMDAR subunit NR1 from PV interneurons does not result in impairment in PPI and spatial working memory[36–38]. To reconcile these discrepancies, couple points should be considered. First, NMDAR hypofunction in PV neurons may not be the sole mechanism contributing to cognitive impairment in the MAM model. Gestational MAM administration selectively interferes with proliferation and

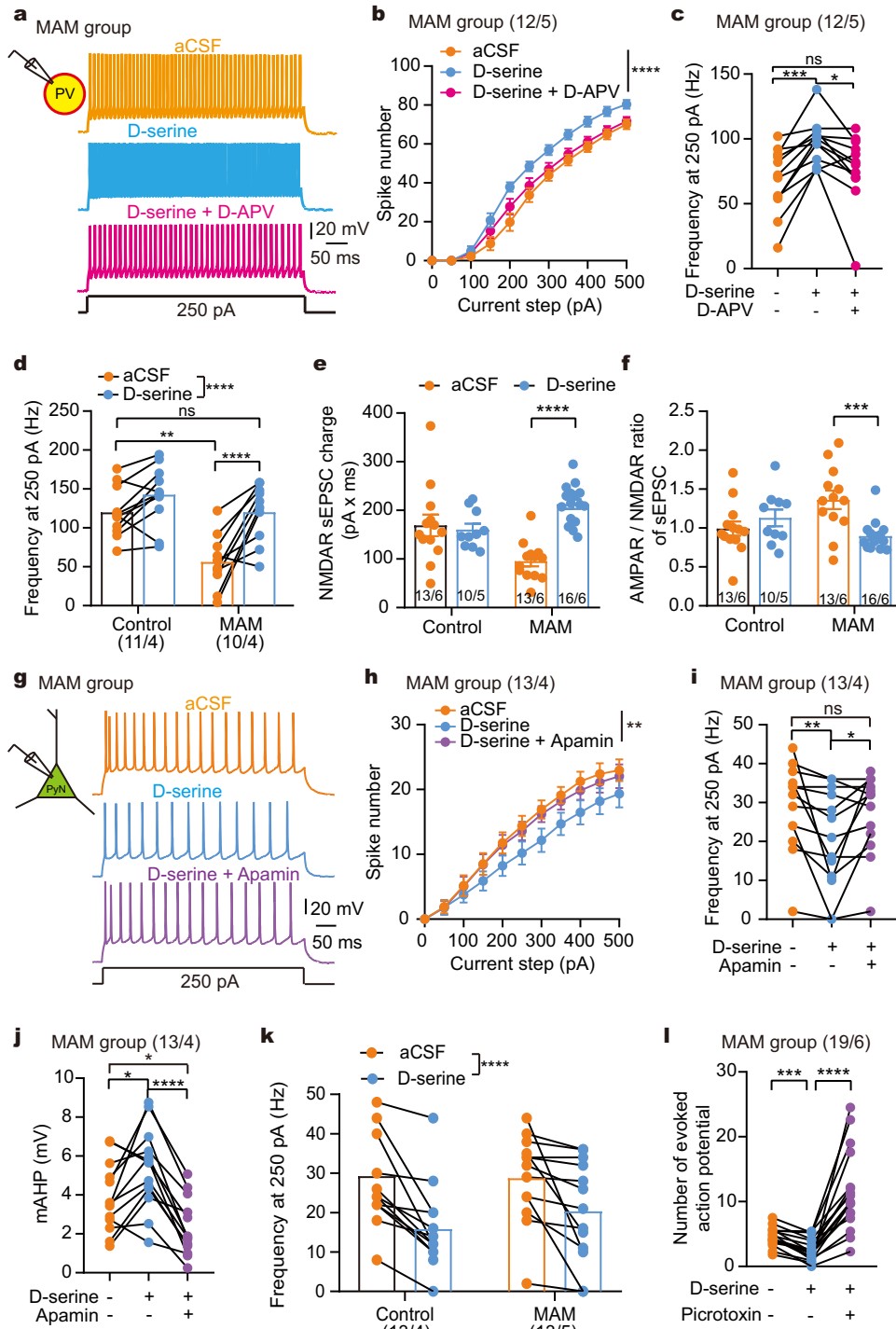

**Fig. 5 | D-serine differentially modulated the intrinsic properties of PV and pyramidal neurons and inhibited synapse-driven pyramidal activity in the ACC of MAM group. a, g** Representative traces of action potential in cingulate PV and pyramidal neurons, respectively, from MAM group upon depolarizing current. **b, c** D-APV reversed the D-serine-induced increase in PV excitability. ****$p < 0.0001$ (**b**, two-way ANOVA: drug $F_{(2, 22)} = 15.38$); *$p = 0.0331$, ***$p = 0.0005$ (**c** Bonferroni's *post hoc* test). **d** D-serine had an overall impact on spike number elicited by depolarizing current in PV neurons (Two-way ANOVA: $F_{(1, 19)} = 29.67$, $p < 0.0001$), and MAM group exhibited distinctive sensitivity to D-serine (****$p < 0.0001$, Bonferroni's *post hoc* test). **e, f** D-serine significantly enhanced NMDAR-mediated sEPSC charge (Two-way ANOVA: ****$p < 0.0001$, Bonferroni's test) and decrease the A/N ratio (***$p = 0.0007$) in the MAM group. **h, i** Apamin countered the inhibitory effect

of D-serine on pyramidal excitability. **$p = 0.0052$ (h, two-way ANOVA: drug $F_{(1.333, 15.99)} = 9.03$); *$p < 0.05$, **$p < 0.01$, ****$p < 0.0001$ (Bonferroni's test). **j** Apamin reversed the effect of D-serine on mAHP in pyramidal neurons. *$p < 0.05$, **$p < 0.01$, ****$p < 0.0001$, Bonferroni's *post hoc* test. **k** D-serine had an overall impact on spike number in pyramidal neurons (Two-way ANOVA: D-serine $F_{(1, 24)} = 23.48$, $p < 0.0001$; prenatal exposure $F_{(1, 24)} = 0.24$, $p = 0.626$). **l** D-serine decreased the average number of eAP in MAM group. ***$p = 0.0002$, ****$p < 0.0001$, Bonferroni's *post hoc* test following one-way ANOVA. Data are shown as mean ± SEM in **b, d–f, h, k**. The first and second N numbers in parentheses or bars correspond to the numbers of recorded neurons and mice, respectively. Source data are provided as a Source Data file.

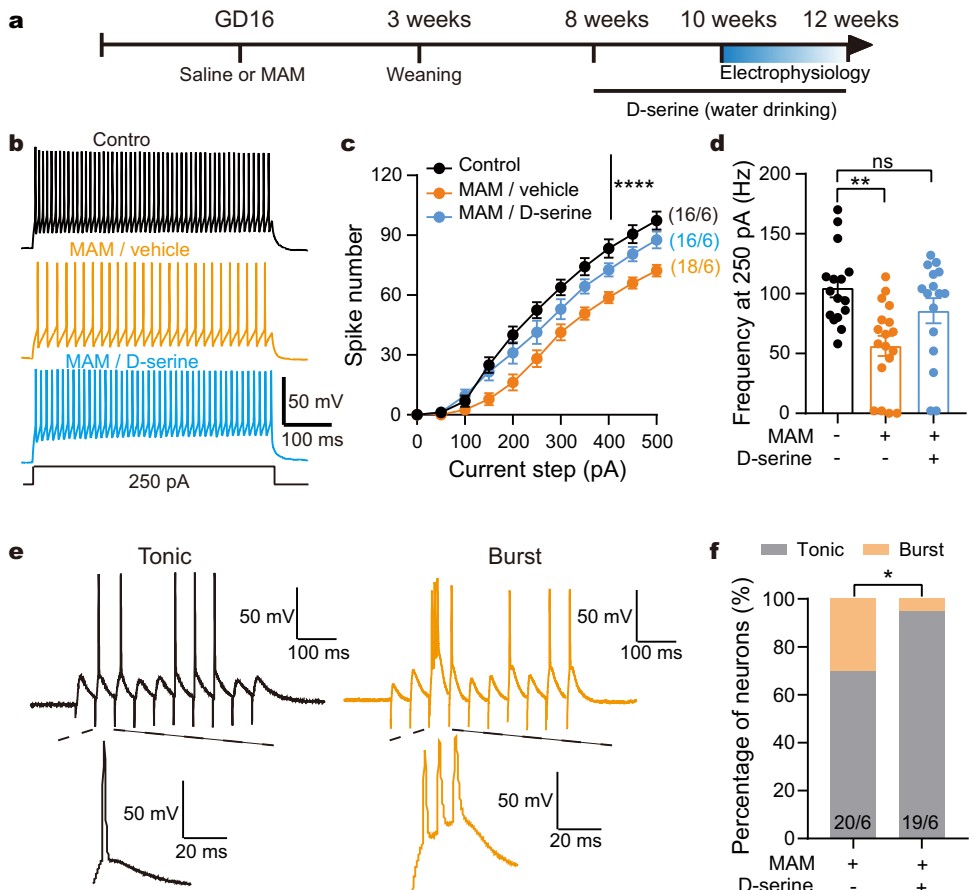

**Fig. 6 | Oral D-serine administration restored the intrinsic properties of PV neurons and reestablished the disrupted E/I balance in the local ACC circuit of the MAM group. a** Diagram of the experimental process. **b** Representative traces of action potential elicited by a depolarizing current step in cingulate PV neuron sampled from control and MAM group or MAM group with systemic D-serine. **c, d** Systemic D-serine restored PV excitability in the MAM group. ****$p < 0.0001$ indicating the impact of drug on spike number in summarized current-firing curves (Two-way ANOVA: $F_{(2, 47)} = 11.28$); **$p = 0.0011$, Bonferroni's *post hoc* test for firing

frequency to 250 ρA depolarizing current (One-way ANOVA: $F_{(2, 47)} = 7.505$). **e** Different patterns of tonic (left) and burst (right) firing during eAP recording in pyramidal neurons. **f** Percentage of neurons exhibiting either tonic or burst firing in all recorded neurons in MAM/vehicle or MAM/D-serine group (two-sided Chi-square test, *$p = 0.0397$). Data are shown as mean ± SEM except **f**. The first and second N numbers in parentheses or bars correspond to the numbers of recorded neurons and mice, respectively. Source data are provided as a Source Data file.

migration of neuronal precursor cells in offspring, leading to a complex and multifaceted prenatal pathogenesis. This suggests that other factors beyond NMDAR hypofunction in PV neurons could contribute to the observed cognitive deficits. Second, while our findings demonstrate that D-serine or the specific correction of PV hypofunction improves cognitive function in the MAM model, this does not preclude the possibility that NMDAR dysfunction in other interneurons may collectively contribute to the pathophysiology of MAM model. The improvement of cognitive function following the correction of PV hypofunction indicates that this strategy has therapeutic potential, but it does not negate the role of NMDAR dysfunction in other cell types.

It is noteworthy that most studies utilizing the MAM model have been conducted in rats[39–41], showcasing the model's stability and reproducibility in generating schizophrenia-like symptoms and PV neuron dysfunctions. Our mouse-based MAM model has also been shown to replicate these findings observed in the rat model. Furthermore, the reliability of the mouse MAM model is corroborated by multiple independent studies, each reporting consistent PPI impairment[25,29,42]. Thus, we posit that the effective mechanisms of D-serine ameliorating cognitive impairment in the MAM mouse model, as discovered in our study, may also be applicable in the more widely used rat model. It is important to acknowledge the potential of sex differences in the pathological alterations caused by prenatal MAM, as

reported by an earlier study reported decreased PV expression and PFC-dependent cognitive deficits exclusively in males[29]. In our study, we did not observe significant sex-dependent differences in PV neuron dysfunction or in the impairment of spatial and recognition memory in the MAM group (Supplementary Table 2). This discrepancy could potentially be attributed to the fact that the neuropathophysiological and behavioral tests we employed were more sensitive in detecting the impact of prenatal MAM across both sexes.

### How does D-serine modulate the excitability of cingulate neurons under control conditions?
The role of NMDAR in regulating neuronal excitability appears to be discrepant and depends on the age and cell type. Directly blocking the NMDAR with an antagonist inhibits the excitability of pyramidal neurons in the hippocampal samples from neonatal mice[43] but results in hyperexcitability of prefrontal inhibitory neurons[44], DRG primary sensory neurons[45], or medium spiny neurons[46] from adolescent or adult mice. Tonic activation of NMDAR by NMDA promotes the excitability of pyramidal neurons in the cortical samples from juvenile rats[47,48], while AICP (a glycine-site superagonist) has the opposite effect on inhibitory neurons of the thalamus[49]. Our results show that activating NMDAR's 'glycine modulation site' with D-serine suppresses the excitability of cingulate pyramidal neurons in an SK (rather than BK) channel-dependent manner. Within the K channel family, both SK and

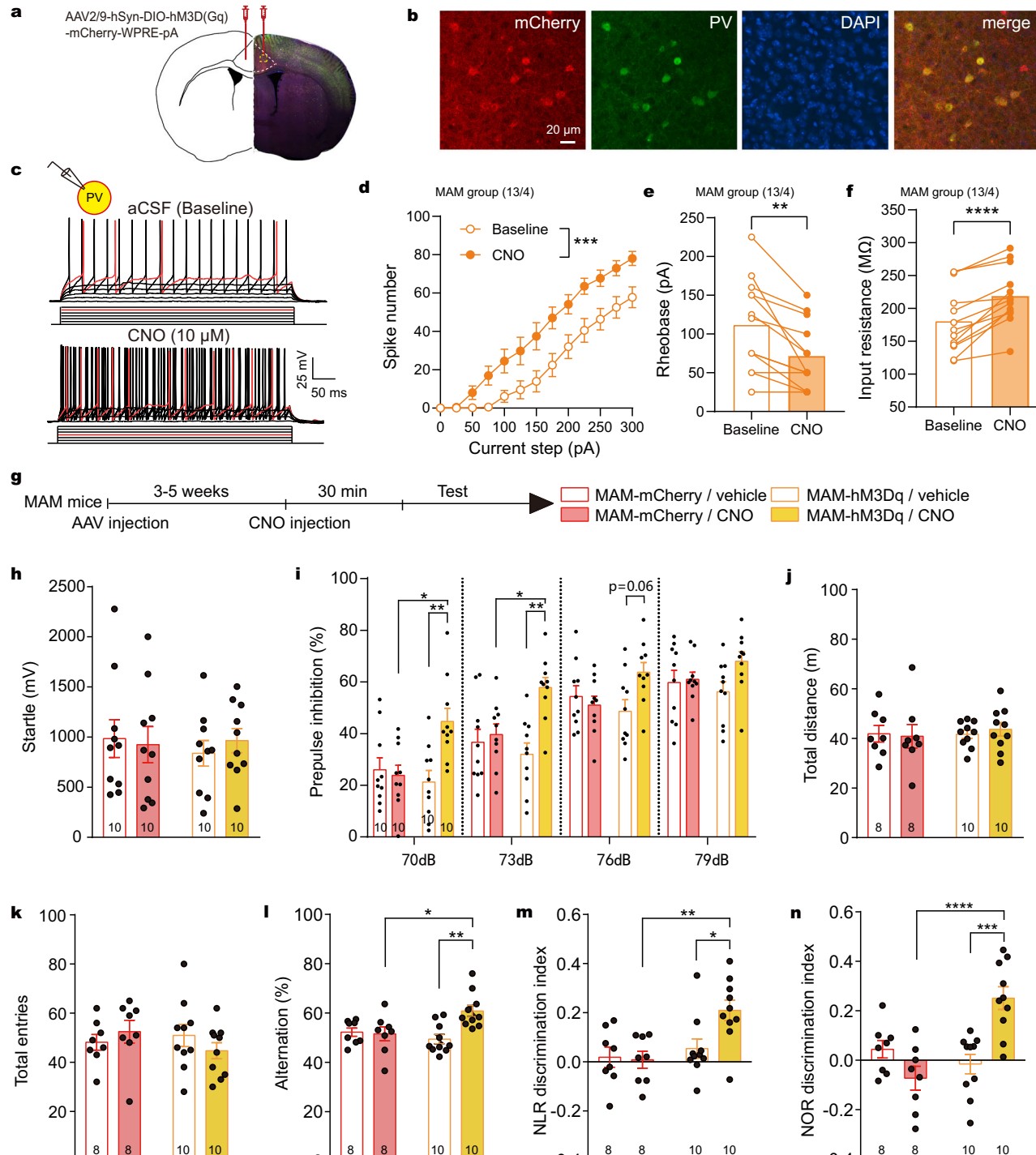

**Fig. 7 | Chemogenetic activation of PV neurons ameliorated cognitive deficits in MAM group. a** Diagram of AAV virus microinjected into the ACC of PV-cre mice. **b** hM3Dq-mCherry was specifically expressed in PV neurons. This experiment was repeated 4 times with similar results. **c** Typical traces of action potentials from the hM3Dq-positive PV neuron pre-/post CNO. **d–f** CNO increased spike number (Two-way ANOVA: $F_{(1, 12)} = 30.80$, ***$p = 0.001$) and input resistance (two-sided paired t-test: $t_{(12)} = 6.187$, ****$p < 0.0001$), but decreased rheobase (two-sided paired t-test: $t_{(12)} = 4.193$, **$p = 0.0012$) of hM3Dq-positive PV neurons. **g** Timeline of AAV injection and CNO/vehicle administration in PV-cre mice experiencing prenatal MAM. **h, i** Activation of PV neurons showed no effect on acoustic startle, but promoted the PPI in MAM group (two-way ANOVA for 70 dB: virus x CNO $F_{(1, 36)} = 7.928$, $p = 0.0078$, virus $F_{(1, 36)} = 3.176$, $p = 0.0831$, CNO $F_{(1, 36)} = 5.443$, $p = 0.0253$; for

73 dB: hM3Dq × CNO $F_{(1, 36)} = 6.761$, $p = 0.0134$, hM3Dq $F_{(1, 36)} = 2.369$, $p = 0.1325$, CNO $F_{(1, 36)} = 10.80$, $p = 0.0023$). **j, k** No change in locomotor activity in open field nor Y-maze total entries upon PV neuron activation (two-way ANOVA). **l–n** Improved spatial working memory (two-way ANOVA: AAV x CNO $F_{(1, 32)} = 7.511$, $p = 0.0099$), and recognition memory (NLR test: AAV × CNO $F_{(1, 32)} = 4.309$, $p = 0.046$; NOR test: AAV × CNO $F_{(1, 32)} = 19.77$, $p < 0.0001$). *$p < 0.05$, **$p < 0.01$, ***$p < 0.001$, ****$p < 0.0001$ Bonferroni's *post hoc* test. Data presented as mean ± SEM. The first and second *N* numbers in parentheses correspond to the numbers of recorded neurons and mice, respectively, while *N* numbers in or below bars indicate the number of mice assigned in behavioral experiments. Source data are provided as a Source Data file. CNO clozapine N-oxide, NLR novel location recognition, NOR novel object recognition.

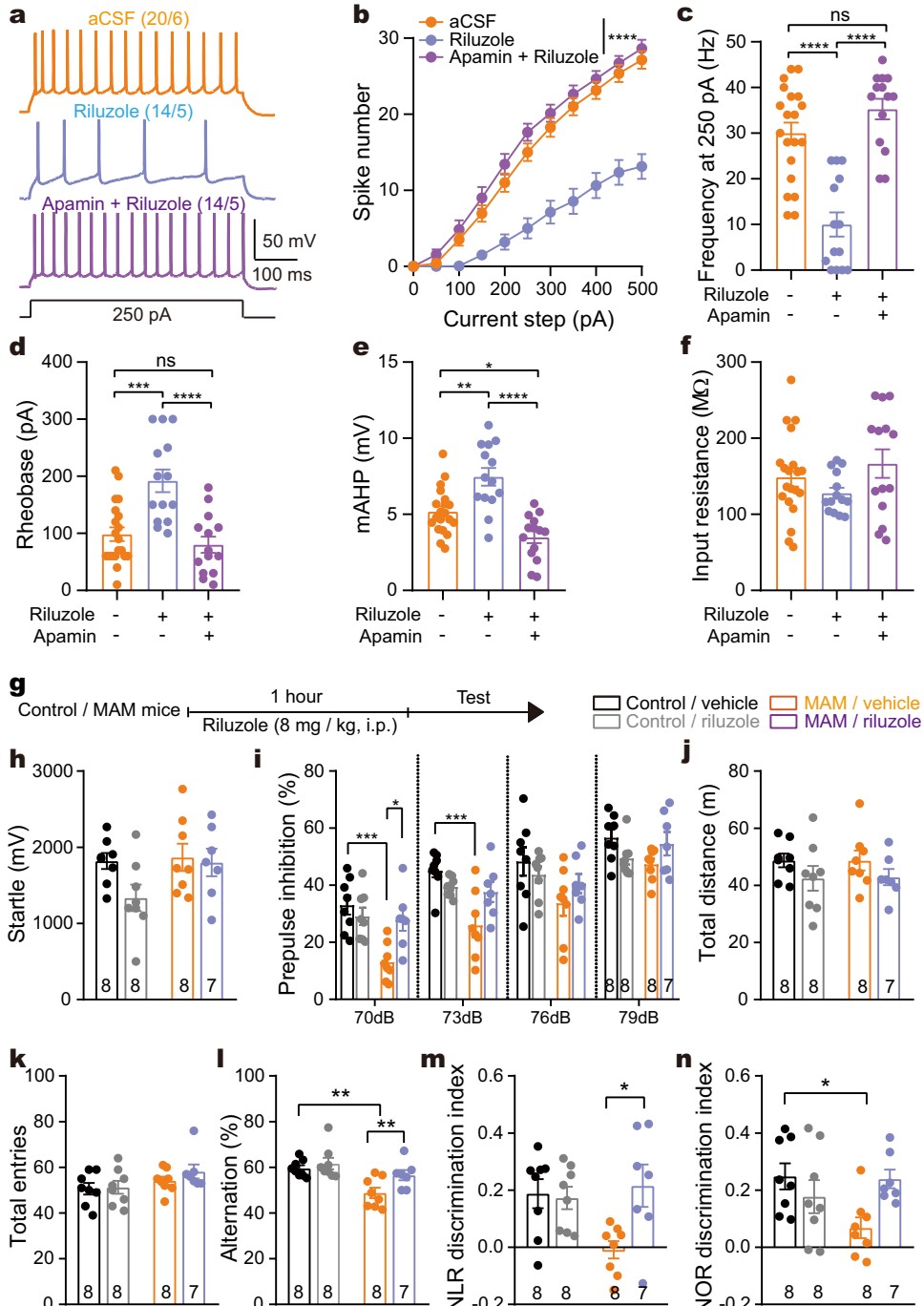

**Fig. 8 | Riluzole ameliorated cognitive deficits in MAM group. a** Typical traces of action potentials elicited by a depolarizing 250 pA current step in pyramidal neuron before and after riluzole or combined riluzole + apamin. **b** Riluzole decreased spike number of pyramidal neurons in the ACC of MAM group. ****$p < 0.0001$ indicating the impact of drug on spike number (Two-way ANOVA: drug $F_{(1.925, 36.58)} = 31.39$). **c–f** Apamin prevented the effect of riluzole on firing frequency to 250 pA depolarizing current (One-way ANOVA: $F_{(2, 45)} = 26.92$, $p < 0.0001$), rheobase ($F_{(2, 45)} = 14.38$, $p < 0.0001$) and mAHP ($F_{(2, 45)} = 18.88$, $p < 0.0001$), but not on input resistance ($F_{(2, 45)} = 1.821$, $p = 0.174$) of pyramidal neurons. *$p = 0.0191$ (**e**), **$p = 0.0012$ (**e**), ***$p < 0.001$, ****$p < 0.0001$, Bonferroni's *post hoc* test. **g** The time course of riluzole administration in MAM mice. **h** Riluzole had no effect on acoustic startle responses (Two-way ANOVA). **i** Riluzole restored the impaired PPI in MAM

group (Two-way ANOVA: *$p = 0.0169$, ***$p = 0.0006$, Bonferroni's *post hoc* test). **j–l** Riluzole had no effect on locomotor activity in OFTs or total entries in Y-maze tests (Two-way ANOVA), but improved spatial working memory in the MAM group (**$p < 0.01$ Bonferroni's *post hoc* test). **m**, **n** Riluzole restored recognition memory in the MAM group (*$p = 0.0247$ for **m** $p = 0.0462$ for **n** Bonferroni's *post hoc* test). Data are shown as mean ± s.e.m. The first and second N numbers in parentheses correspond to the numbers of recorded neurons and mice, respectively, while N numbers in or below bars indicate the number of mice assigned in behavioral experiments. Source data are provided as a Source Data file. mAHP medium afterhyperpolarization, NLR novel location recognition, NOR novel object recognition.

BK channels are activated in a $Ca^{2+}$-dependent manner, mediating the medium and fast AHPs, respectively, and playing a role in long hyperpolarizing pauses that increase the amplitude and duration of AHPs[50]. In addition to voltage-gated $Ca^{2+}$ channels, NMDAR-mediated $Ca^{2+}$ influx is also an important source for activating SK and BK channels[51]. Recent studies have demonstrated the functional coupling of NMDAR-SK or NMDAR-BK results in hypoexcitability in barrel cortex pyramidal neurons[52] or hypothalamic magnocellular neurosecretory neurons[53]. Notably, our results reveal that BK channels are less sensitive to D-serine than SK and are not involved in D-serine-induced inhibition of cingulate pyramidal neuron excitability in adult mice (Supplementary Fig. 1). This result is in line with the role of BK in regulating pyramidal firing, which limits spike broadening without affecting firing frequency[54].

In PV neurons, there are strong reasons to believe that NMDAR has a weak contribution to directly regulating synaptic activation due to specific neurochemical and biophysical properties. As demonstrated in our recorded cells, PV neurons can generate high firing frequency with quite short duration, but NMDAR exhibits slow kinetics, which limits its participation in PV neuron excitation[16]. On the other hand, parvalbumin and calbindin, which are highly expressed in PV neurons, are powerful $Ca^{2+}$-binding proteins that provide fast and sufficient chelation for NMDAR-mediated $Ca^{2+}$ influx[17,18]. Thus, unlike in pyramidal neurons, SK and BK channels have less probability to be activated by NMDAR-mediated $Ca^{2+}$ influx in PV neurons[55]. Our recordings demonstrate that unlike in pyramidal neurons, D-serine does not facilitate NMDAR-mediated current in PV neurons sampled from the control group (Fig. 5e, f). In line with this finding, the intrinsic excitability of PV neurons under baseline conditions or during D-serine incubation are independent of the NMDAR (Figs. 1n and 5d). This implies that under physiologically relevant conditions, NMDAR does not play a role in PV neuron excitability, and D-serine does not change this relationship. Typically, pyramidal neurons receive inhibitory input from PV neurons proportional to the excitatory input they receive, thus maintaining E/I balance[56]. This may explain why D-serine does not affect synapse-driven pyramidal activity in the control group (Fig. 1o–q).

## How does D-serine affect dysfunctional cingulate neurons in the neurodevelopmental model for schizophrenia?

The MAM group exhibits the impaired spatial and recognition memory, as well as deficits in sensorimotor gating, consistent with previous findings on cognitive dysfunction in this model[29,30,39]. Although pyramidal neurons in the ACC of the MAM group retain functional NMDAR and normal intrinsic excitability, PV neurons display hypoexcitability and a reduction in NMDAR-mediated current. Increased $K^+$ channel densities, as implied by reduced membrane resistance and RMP, might be related to PV hypoexcitability. Recordings from pyramidal neurons in the MAM group show a higher probability of burst firing in response to synaptic input and a lower probability of synaptic GABA release relative to the control group. This might be attributed to the reduced inhibitory output from PV to pyramidal neurons (Fig. 4a–h). These results support the previous reports of deficient inhibitory control in local cortical circuits due to PV neuron hypofunction, which has been highlighted as a key point of pathological convergence in schizophrenia[57]. Nevertheless, it is important to consider the potential impact of other variables that may contribute to the reduced inhibitory input from PV neurons, including cortical atrophy or a decrement in PV neuron density[29,58].

Compared to the control group, D-serine more remarkably increases PV excitability in the MAM group, an effect mediated by NMDAR. Furthermore, D-serine facilitates NMDAR-mediated current in PV neurons sampled from the MAM group but not from the control group. We suppose that chronic PV hypoexcitability in the ACC of the MAM group attenuates the $Ca^{2+}$ chelation or scales up the sensitivity of

intracellular signaling cascades sensing $Ca^{2+}$ influx to maintain functional stability via homeostatic plasticity[59], which amplifies the promoting effect of D-serine on PV excitability. The facilitation of PV excitability by D-serine shifts the E/I balance to 'I' as evidenced by diminished synapse-driven pyramidal activity (Figs. 5l and 6f), revealing the overall inhibitory effect of D-serine on cingulate microcircuits in the MAM group.

The MAM group treated with D-serine in their drinking water exhibits a partial restoration of PV excitability and less probability of burst firing during stimulus trains in the ACC. These findings suggest that the therapeutic effects of D-serine on cognitive function are mediated, at least in part, by the reestablished E/I balance in cortical circuits. It is possible that these neuroadaptive changes are not only a result of D-serine's primary effects, but also of lasting modifications in synaptic plasticity, intracellular signaling, and neurocircuitry.

Taken together, D-serine might inhibit cingulate pyramidal neurons in the MAM group at least in two ways. First, when NMDARs of PV interneurons are stimulated directly by synaptic glutamate, D-serine promotes inhibitory input to pyramidal neurons. Second, tonic activation of NMDA receptors by ambient glutamate in conjunction with gliotransmitter (e.g., D-serine or glycine) preferentially inhibits pyramidal excitability via triggering SK channel opening. Notably, the latter conclusion was drawn because D-serine exhibited apamin-sensitive effects on the membrane properties. Apamin has been recognized as a specific inhibitor of SK channels due to its ability to block these channels at nanomolar or even subnanomolar concentrations, with insignificant effects on other molecular targets[60]. However, it may still have unforeseen actions on other molecular targets in our experimental setting. Therefore, this study does not establish a causal relationship, but rather a correlation between D-serine's effect and SK channels activation.

## Reconstitution inhibitory control via directly strengthening PV neurons or activating SK channels is sufficient to improve cognitive function in the MAM model

The above results show D-serine boosted the intrinsic excitability of PV neurons but weaken that of pyramidal neurons, via different cellular mechanism in the MAM group. However, it could not exclude the role of non-PV neurons in cognitive improvement. Through a chemogenetics strategy, we found that restoring inhibitory control to cortical pyramidal neurons via selectively strengthening PV neurons was sufficient to improve cognition in the MAM group. Furthermore, in control group, an enhancement of PV excitation in the ACC does not improve cognitive function (Supplementary Fig. 5), suggesting that the behavioral improvements observed in the MAM model are specifically due to correction of PV hypofunction.

Another interesting finding was that systemic riluzole treatment was an effective way to improve cognitive function in this model. Riluzole has been used to slow the course of amyotrophic lateral sclerosis via protecting neuron against excitotoxicity. Although riluzole targets multiple cation channels, its affinity for SK channels is relatively high[61], and it is the only drug that can activate SK channels in clinical practice. Consistent with results on other type of neurons[62], riluzole significantly promoted mAHP and SK-dependently inhibited cingulate pyramidal excitability. Notably, a clinical imaging research indicated that riluzole decreased glutamate metabolism and increased ACC-PFC connectivity in treatment-resistant schizophrenia[63]. These results imply that SK channel might be potential and promising target for improving cognitive dysfunction in schizophrenia.

In conclusion, our findings unveiled a novel dual mechanism of D-serine in cognitive improvement using a neurodevelopmental model for schizophrenia. That is D-serine reconstituted synaptic and intrinsic inhibitory control of cingulate pyramidal neurons via facilitating PV excitability and activating pyramidal SK channels,

respectively. Either directly strengthening PV neuron activity or activating SK channels is sufficient to improve cognitive function in this model. These findings suggest that restoration of inhibitory control over cortical microcircuit may be an efficacious strategy for the treatment of cognitive impairment seen in conditions where NMDAR deficits may be a pathological feature.

## Methods

### Animals

All experiments were conducted in line with the National Institutes of Health Guide for the Care and Use of Laboratory Animals, which were approved by the Animal Care and Use Committees of Ningbo University. PV-Cre mice (Jackson laboratory#008069; C57BL/6J background) express Cre recombinase under the control of the mouse parvalbumin (PV) promoter, steering Cre expression specifically to parvalbumin expressing cells. Time-mated C57BL/6J or PV-Cre (+) dams were injected intraperitoneally (i.p.) with MAM (26 mg/kg; Wako Pure Chemical Industries #136-16303) or vehicle (0.9% NaCl) on gestational day (GD) 16[30]. On postnatal day (PD) 21, male and female pups were weaned and separately housed in groups of 3–5 with littermates who are the same treatment group until adulthood. Adult mice, both male and female (the numbers of two sexes were nearly equal in each group), were used in the experiment. To avoid litter effects, in each experimental group, no more than two pups from a single litter were used. All mice were housed under standard conditions at 22 °C with 55% humidity and a 12 h light–dark cycle with free access to food and water. All experiments were performed and analyzed by experimenters who were blind to both the genotype of the animals and the drug treatments.

### Stereotaxic surgeries

Mice were anesthetized with sodium pentobarbital (80 mg/kg, i.p.) and placed onto a stereotaxic device. The adeno-associated virus (AAV) vectors were infused bilaterally into the ACC region (AP: +1.0 mm, DV: −2.0 mm, ML: ±0.35). A volume of 0.3 μL was infused at a speed of 0.1 μL/min using a glass needle connected by Tygon tubing to a 10 μL Hamilton needle syringe by a Syringe Pump Controller. When the injection was done, the needle was left standing for 5 min to prevent solution backflow. Ketoprofen (5 mg/kg, s.c.) was administered as an analgesic before surgery as well as 24 h after surgery. After surgery, experiments were conducted 3 weeks post injection.

### Chemogenetic/optogenetic manipulation of PV interneurons

The AAV vectors (AAV2/9-hSyn-DIO-mCherry, AAV2/9-hSyn-DIO-hM3D(Gq)-mCherry-WPRE-pA, AAV2/9-hEF1a-DIO-hChR2(H134R)-mCherry) were purchased from Taitool Bioscience Co., Ltd (Shanghai, China). For chemogenetic manipulation, mice were intraperitoneally injected with CNO dissolved in 0.9% saline or with 0.9% saline only. Thirty minutes before each behavioral test, mice received Clozapine N-oxide (CNO, 2 mg/kg) for acute treatment. When eletrophysiological recording in vitro, CNO was diluted in aCSF to final concentration of 10 μM. For optogenetic manipulation, PV neurons were activated by a blue laser (473 nm) with 10 mW at 20 Hz.

### Supplementation of drugs

MAM mice and control mice were given D-serine (600 mg/L, Sigma-Aldrich, St. Louis, MO, USA) in their drinking water for two weeks before and during the behavioral tests, following a protocol based on previous research[28]. On average, a mouse consumed approximately 5 mL of water per day, resulting in a daily dose of approximately 100 mg/kg of D-serine[28]. Notably, we observed minimal variability in water intake among the mice. For systemic Riluzole application (8 mg/kg, i.p.), a daily preparation was made using saline containing 2% Tween-80 (P6474, Sigma) and mixed at room temperature for at least 1 h before use.

### Behavioral tests

All behavioral experiments were performed in adult mice (8–12 weeks old) between 7:00 a.m. and 18:00 p.m., day phase of mice. Prior to the behavioral test, mice were habituated to a room and to a single experimenter by handling in the behavioral room for 5 min. After each behavioral stage, the apparatus was thoroughly cleaned with 75% ethanol to prevent olfactory cue bias.

### Pre-pulse inhibition (PPI) test

Pre-pulse inhibition test was performed to assess sensorimotor gating by acoustic startle response system (AniLab Scientific Instruments Co., Ltd, China). In this experiment, mice were placed in a Plexiglas rectangular tube on a platform with accelerometer and highly sensitive motion sensor, and the platform were in a sound attenuated, ventilated cubicle, with a speaker above the platform. Acoustic startle response of mice was recorded by LabState ASR Software. Before the test, the mice were acclimated to the startle system for 10 min daily for 3 days with 65 dB background white noise, but not acoustic stimulus. After habituation, PPI was measured in a 40 min period with 66 randomize trials. At the beginning of each session, mice were placed into the tube for a 5 min acclimatization period like the last three days and background white noise ran through the whole 40 min period. After acclimatization, there were 66 trials presented at randomized inter-trial intervals (25–40 s). The first five and last five trials were 120 dB startle pulse, which were carried out to measure habituation to the startle stimulus. The middle 56 trials were presented in a pseudorandom order: the mode of 40 trails consists of prepulse (none, 70, 73, 76, 79 dB) and pulse (120 dB), and each prepulse-pulse occurred 8 times; the mode of remaining 16 trials was only prepulse (70, 73, 76, 79 dB), and each prepulse occurred 4 times. The duration of any pulse is 40 ms and the pulse appeared 100 ms after the prepulse. Percentage of prepulse inhibition for each mouse at each prepulse was calculated by the formula: PPI (%) = $(1 - A/B) \times 100\%$, where A is the average startle response on trials in which there was a prepulse and B is the average startle response on the trials in which the only pulse was occurred (excluding the first five and last five trials).

### Open field test (OFT)

OFTs were performed to test locomotor activity by a white box (40 cm L × 40 cm W × 40 cm H) made of Plexiglas plate. Mice were placed in the center of the bottom of the box to explore freely for 10 min and the total distance traveled was measured using ANY-maze software (Stoelting, US).

### Y-maze

Y-maze tests were performed to assess short-term spatial working memory by spontaneous alternation. The device is a three-arm horizontal maze (40 cm L and 10 cm W with 25 cm H) in which the angle between each of the two adjacent arm is 120°. Mice were placed at the junction of the three arms and allowed to explore freely for 8 min. The total entries and sequence of arms were recorded. The precent alternations, reflecting spatial working memory ability, were defined as the proportion of arm choices that differed from the last two choices.

### Novel location recognition (NLR) and Novel object recognition (NOR)

NLR and NOR tests were performed to test recognition memory in mice. In the NLR test, the device includes an open field box (25 cm L × 25 cm W × 25 cm H, one of whose walls is specially marked) and two identical objects. Mice were acclimated to the open field box for three days, 10 min a day. During training, 24 h after the last acclimatization, mice were allowed to explore two identical objects for 10 min. Investigation time for each object was measured. One hour after training for test, one of the objects was picked up and placed diagonally opposite the other and mice were allowed again to explore two objects for

5 min. The investigation time for each object was measured again. Object exploration time was measured for each case in which a mouse's nose touched the object or was oriented toward the object and came within 2 cm of it. The NLR discrimination index, reflecting spatial recognition memory ability, was defined as (novel location investigation time – familiar location investigation time)/(novel location investigation time + familiar location investigation). The device for NOR test includes an open field box (25 cm L × 25 cm W × 25 cm H) and two objects of different shapes but of the same material. The acclimating and training phases are the same as NLR test in the first four days. On the fifth day (24 h after the training phases), one of the objects was replaced with a new object, mice were allowed to explore two different objects for 5 min. The data is recorded in the same way as NLR test. The NOR discrimination index, reflecting recognition memory, was defined as (novel object investigation time – familiar object investigation time) / (novel object investigation time + familiar object investigation time).

### Electrophysiology

Coronal brain slices containing ACC were prepared for whole-cell recordings from control and MAM groups during PD 70–84. Mice were anesthetized with sodium pentobarbital (80 mg/kg, i.p.) and decapitated, and then brains were dissected quickly and placed in an ice-cold solution containing below substances (in mM): 75 sucrose, 87 NaCl, 3 KCl, 1.5 $CaCl_2$, 1.3 $MgCl_2$, 1 $NaH_2PO_4$, 26 $NaHCO_3$, 20 glucose equilibrated with 95% $O_2$–5% $CO_2$. Coronal brain slices (220 μm thickness) were prepared with a vibratome (Leica VT1200S, Leica Microsystems, Germany), and then incubated in a chamber with artificial cerebrospinal fluid (aCSF) containing below substances (in mM): 124 NaCl, 3 KCl, 1 $NaH_2PO_4$, 1.3 $MgCl_2$, 2 $CaCl_2$, 26 $NaHCO_3$, and 20 glucose, 295–305 mOsm, equilibrated at 32 °C with 95% $O_2$–5% $CO_2$. Slices were incubated for at least 1 h before recording. Following incubation, the slices were transferred to a recording chamber, where the submerged slices were perfused with aCSF (32 °C) saturated with mixed gas at a flow rate of 2 mL per min. Standard recordings were made using Multiclamp 700B amplifier and Digidata 1550B (Molecular Devices, Axon Instruments, CA, USA) for data acquisition. Vertical two stages puller (PC-10, NARISHIGE) was used to make glass electrodes (3IN thinwall GL1.5 OD/1.12 ID, TW150–3, WPI) into pipettes with resistance between 1.5 and 2 mOhm when filled with internal solution.

For whole-cell recordings, an average of 3 neuronal cells (ranging from 1 to 5) were sampled per animal, with the majority falling within the range of 2–4 neurons. Pyramidal neurons from the layer V were visually identified based on their shape and prominent apical dendrite. PV neurons were identified by mCherry fluorescently. In voltage clamp recording, patch pipettes were filled with an intracellular solution containing (in mM): 110 cesium methylsulfate, 15 CsCl, 4 Mg-ATP, 0.3 $Na_2$-GTP, 0.5 EGTA, 10 HEPES, 4 QX-314, 5 Phosphocreatine-$Na_2$, pH 7.2–7.4, 270–280 mOsm. sEPSCs and sIPSCs were recorded using the cesium-based internal solution. Picrotoxin (100 μM, P1675, Sigma) was added in the regular aCSF when sEPSCs were recorded with cells being held at −70 mV. sIPSCs were recorded with cells being held at 0 mV. In optogenetic stimulation experiments, the procedure is generally as reported in the literature[64]. The recorded cell was held at −70 mV in voltage clamp, and the response to a current evoked by a 5-ms pulse of blue laser (473 nm) was recorded. For presynaptic release probability experiments, four 5-ms pulse were applied at 20 Hz and the response to each pulse was analyzed. The AMPAR/NMDAR ratio was measured from the evoked EPSCs while holding PV interneurons at +40 mV. The AMPAR EPSCs were isolated after bath application of the NMDAR antagonist D-2-amino-5-phosphonovaleric acid (D-APV, 100 μM, 14539, Cayman). The NMDAR EPSCs were obtained by digital subtraction of the AMPAR EPSCs from the dual (AMPAR + NMDAR) EPSCs.

AMPAR- or NMDAR-mediated sEPSCs were detected with digitally designed templates (Molecular Devices). The sEPSC charge was computed by the following formula: sEPSC charge = current (ρA) × time (ms). For dual sEPSCs, a template with rise and decay times of 3 and 150 ms, respectively, was used. A lower-amplitude threshold of 16 ρA was applied. AMPAR sEPSCs at +40 mV were isolated in the presence of 100 μM D-APV with a template with rise and decay times of 1.2 and 4 ms, respectively. A lower-amplitude threshold of 9 ρA was applied. The NMDAR sEPSCs were obtained by the following formula: NMDAR sEPSCs = dual (AMPAR + NMDAR) sEPSCs – AMPAR sEPSCs.

For current-clamp recordings, a K-based internal solution was used (in mM): 120 K-gluconate, 10 KCl, 10 HEPES, 0.5 EGTA, 4 Mg-ATP, and 0.3 Na-GTP, 5 Phosphocreatine-$Na_2$, pH 7.2–7.4, 270–280 mOsm. Before current-clamp recording, series resistance was monitored and fully compensated using the bridge balance. A current step protocol (from −200 to 500 ρA, with a 50 ρA increment; inter-pulse interval, 15 s) was then run for at least three runs. RMP and input resistance were recorded after breaking in, by a 500-ms duration −100 ρA current injection. The rheobase was the minimum current injected (500 ms, 1 Hz) that generated action potentials in 50% of the cases. The threshold was estimated as the point when the slope of rising membrane potential exceeds 50 mV ms$^{-1}$ and valued from the first current injected step. The amplitude was the voltage increment between the resting level and spike voltage peak. The amplitude of mAHP was measured as the peak negative membrane (from baseline) following a single action potential[65,66]. The inhibition rate of mAHP amplitude by apamin was used to evaluate the function of SK channel[67]. The half-width was valued from the 200-ρA current injected step, and neurons without action potential in this step will not be counted. The value of the decay time of the action potential was determined as the width of the spike at half-maximal amplitude. To measure the evoked action potential (eAP) from ACC pyramidal neurons, a bipolar tungsten electrode (FHC, Inc., United States) was placed 150–200 μm away from the recording neuron. Stimulus strength was adjusted to 40–60% of the maximal eAP response. Synapse-driven pyramidal activity was determined with trains of synaptic stimulation (10 trains were applied at 40 s intervals; each train consisted of 10 pulses at 20 Hz) with the pyramidal neurons at RMP in the absence of synaptic blockers. Only cells with stable RMP and stable evoked action potential firing were included. Series resistance was normally less than 20 MΩ and recordings exceeding 20% change in series resistance were terminated. All holding potentials were corrected for liquid junction potential. Data were low-pass filtered at 1 kHz and digitized at a sampling frequency of 10 kHz for on-line and later off-line analysis (Clampfit 10.7; Molecular Devices, CA, USA). Drugs of D-serine (10 μM, S4250, Sigma), riluzole (2 μM, HY-B0211, MCE), apamin (0.2 μM, HY-P0256, MCE) and paxilline (1 μM, HY-N6778, MCE) were added in the regular aCSF to activate NMDAR, activate SK, block SK and block BK channel, respectively.

All chemicals used in the patch clamp were purchased from Sinopharm Chemical Reagent Co., Ltd, except as noted.

### Immunofluorescent staining and quantitative analysis

Brains were isolated and fixed in 4% paraformaldehyde (PFA) for 24 h and then cryoprotected in 30% sucrose solution in PBS 1× for additional 2 days. After fixation and cryoprotection procedures, brains were cut using Leica Cryostat in 30 μm thickness. Sections were transferred into a blocking solution containing 0.1% Triton X-100, 10% goat serum in PBS 1× for 1 h at RT (room temperature). Then, sections were incubated at 4 °C overnight with the primary antibody, mouse anti-Parvalbumin (BM1339; BOSTER, Wuhan, China; 1:300) diluted in PBS 1×, 0.1% Triton X-100, and 10% goat serum. After washing with PBS 1× for 1 h, the sections were incubated with secondary antibodies (SA00013-1; Proteintech, Wuhan, China; 1:500) diluted in PBS 1× for 2 h at RT. Afterward, sections were washed by PBS 1× thoroughly for 1 h and mounted with an antifading medium (Solarbio, Beijing, China). For quantitative analysis of PV staining, we counted all labeled cells within each section, covering 100% of the sample area in the different brain area and layers.

## Statistical analysis

GraphPad Prism version 8 (GraphPad Software, San Diego, CA, United States) was used to conduct statistical analyses. An assessment of the normality of data was a prerequisite for choosing parametric or non-parametric testing. Unpaired or paired student's t-test was used to compare two groups. One or two-way ANOVA followed by Bonferroni's *post hoc* test was used for assessing effects within more than two groups in electrophysiological and behavioral experiments. F distribution has two parameters: degrees of freedom in the numerator (DFn) and the denominator (DFd), which were represented as $F_{(DFn, DFd)}$. When the data was normally distributed, the results were presented as mean ± SEM. $p < 0.05$ was considered as statistically significant for all results. Supplementary Table 1 shows the number of neurons and mice used in each group, and a breakdown of the number of male and female mice per group. Sex-based analysis was summarized in Supplementary Table 2.

## Reporting summary

Further information on research design is available in the Nature Portfolio Reporting Summary linked to this article.

## Data availability

All relevant data generated in this study are provided in Source Data file. Source data are provided with this paper.

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

## Acknowledgements

This research was supported by National Natural Science Foundation of China (32171017 and 31571094: H.W.S.; 32201322: X.Q.Z.), the Young Doctor Innovation Research Program of Ningbo (2022J083: X.Q.Z.). We thank Di-Seng Mei from the core facility platform of Ningbo University School of Medicine for their technical support.

## Author contributions

H.W.S. and X.Q.Z. were responsible for the overall experimental design. X.Q.Z. L.X. and X.Y.Z. performed the behavioral tests. X.Q.Z. and Z.H.T. performed the whole-cell electrophysiology experiments with

supervision from H.W.S. L.X. Z.P.Y. and Z.C.W. injected the virus for chemogenetic or optogenetic manipulation. Y.B.D. and Q.S. perfused mice and performed the immunofluorescence staining. X.Q.Z. L.X. and X.Y.Z. performed the data analysis. H.W.S. and X.Q.Z. outlined and wrote the manuscript, which was reviewed by all authors.

## Competing interests

The authors declare no competing interests.
