## [Peer Review File · Nature Communications]

D-serine reconstitutes synaptic and intrinsic inhibitory control of pyramidal neurons in a neurodevelopmental mouse model for schizophreniaREVIEWER COMMENTS

Reviewer #1 (Remarks to the Author):

The paper by Zhang et al. investigates the effects of D-serine in reversing the effects of prenatal MAM treatment. This is a very good and comprehensive paper on the topic. The MAM model is an excellent neurodevelopment model of schizophrenia that allows the detailed investigation of mechanisms, as performed here. However, it cannot be published in the current form. There are methodological/presentation details that need attention and/or elaboration. Furthermore, careful attention should be paid by the authors in the English language. There are several syntactical/grammatical issues and the authors should carefully improve the writing of the paper.

Major issues

English language: past and present tense are intermixed in the introduction. I think the use of present tense should mostly be used in the introduction.

I don't really agree with the results on the mAHP. The mAHP after a single AP is usually not very accurate. The function of SK channels (including mAHP and sAHP) is much better observed and studied after a burst of 3-6APs, usually given following a 50ms step-pulse (see several papers on that such as Matthews EA, Linardakis JM, Disterhoft JF (2009) The fast and slow afterhyperpolarizations are differentially modulated in hippocampal neurons by aging and learning. *The Journal of neuroscience: the official journal of the Society for Neuroscience* 29:4750-4755. Or 1.Liebmann, L. et al. Differential Effects of Corticosterone on the Slow Afterhyperpolarization in the Basolateral Amygdala and CA1 Region: Possible Role of Calcium Channel Subunits. *Journal of Neurophysiology* 99, 958-968 (2008).) On the same point, how reduced mAHP result in reduced excitability? It seems contradictory to me (unless I'm missing something). In most cases and previous studies, reduces the mAHP increased excitability (see references above) Details on how the different electrophysiological properties were analysed should be provided (input resistance, mAHP, etc).

In Figure 4, description of the h panel is missing. I assume it indicates sIPSCs following laser activation of PV interneurons. In this case, representative traces from both control and MAM mice should be shown in cases without ontogenetic activation and with ontogenetic activation.

In Figure 5, What is the difference between panels a, b and c in this figure and Figure 1? I guess this is from animals that received D-serine in the water (i.e. systemic D-serine and not in the bath). This should be indicated in the figure legend, and possibly even in the figure.

With regards to reporting statistical results, it is important to identify in each experiment the number of neurons and the number of mice used in that experiment. Some of the numbers of the F metrics are extremely high, making me think that they refer to individual action potentials. This is not correct. The authors should average the properties of APs per cell and then perform the statistics on the different cells, always indicating the number of mice used per group as well. To better illustrate the numbers of mice used across experiments, a table with the numbers of mice/cells for each experiment should be provided.

Minor issues

D-serine in the water: Were the mice monitored on how much water they were drinking? Was there variability among the mice?

Ln. 477 - NOL should be NOR

Ln 500 - when filled with internal. The word solution should be added.

Were recordings targeted in layer 2, layer 5 or both neurons?

How were the brain slices subjected to immunofluorescence imaged? Besides the zoomed in pictures in figure 6, larger whole slice images should also be provided. This is also true for the optogenetics experiments.

Figure 1f: is this voltage-clamp or current-clamp. The trace looks like current-clamp but the y-axis is indicated in pA.

Reviewer #2 (Remarks to the Author):

MS 390694 Comments to the authors

In light of clinical findings supporting NMDA receptor co-agonist D-serine as a novel therapeutic in the treatment of cognitive domains of schizophrenia, Zhang et al. sought to identify the cellular basis for these effects in the MAM mouse model. Given that PV interneuron and NMDA receptor hypofunction in the prefrontal cortex is a common pathological feature of schizophrenia, the authors conducted whole-cell electrophysiological recordings in PV and pyramidal neurons to determine the influence of D-serine upon the biophysical properties of these cell types. Additionally, behavioral assays were conducted in the MAM model to assess the efficacy of D-serine in treating cognitive deficits associated with schizophrenia. The authors reported some interesting and novel findings. For example, D-serine was capable of rescuing some behavioral deficits in MAM mice. Bath application of D-serine restored the dysregulated excitability of PV and pyramidal neurons in the ACC of MAM mice, and D-serine-mediated excitability regulation in pyramidal neurons was mediated by SK channel activation. This study thus revealed that treatment with D-serine serves to regulate E/I balance via opposing mechanisms between two relevant cell types. The data presented are well-organized and are generally convincing. However, the impact of these findings is limited by the lack of investigation into the mechanisms by which D-serine facilitates biophysical property changes in PV neurons. There are also some controversial issues with the mice MAM model.

Major Concerns

1. First, since the first MAM model was reported by Moore et al. 2006, the general consensus is that MAM administration in rats GD17 produced a relatively stable replication of the pre-clinical symptoms of schizophrenia. In contrast, similar behavioral and physiological changes failed to be reproduced in mice. There were several attempts to develop the MAM model in mice (Chalkiadaki et al, 2019; Hou et al, 2018), including the authors' group (Zhang et al 2021), in recent years. However, considerable variabilities were reported. For example, the dose and timing of MAM administration produced distinct morphological and physiological changes. There were also different behavioral deficits in males vs. females. These variabilities raise a serious concern that have haunted the field for years. The authors did not address this concern. In particular, the authors did not make it clear whether both male and female offspring were used in the study despite the reported sex differences in morphology, physiology, and behaviors. The authors also did not discuss whether these findings in the mice can be reproduced in the more popular rat MAM model. This concern thus seriously compromises the implication of the study.
2. While the authors set out to investigate NMDA hypofunction in PV neurons and the therapeutic potential of D-serine, the experiments did not directly link functional improvements in PV neuron excitability with NMDA function, following D-serine application. This is true of D-serine application in both control and MAM contexts.
3. The therapeutic relevance of D-serine is limited in this study, given that the compound was only applied via bath administration during electrophysiological recordings. Thus, it remains unclear if the restoration in PV excitability would recapitulate systemic administration of D-serine were to be tested. Thus, it is unclear if the behavioral effects of D-serine are due to the restoration of E/I balance in cortical circuits, or some other mechanism.
4. While it is an interesting finding that D-serine administration induces opposing biophysical property changes in pyramidal and PV neurons, the authors did not attempt to further investigate, for example, the precise mechanism by which NMDA receptor co-agonism by D-serine lead to increased excitability in PV neurons. This is problematic, considering the title of the manuscript.
5. Chemogenetic experiments shown in the study did not seem relevant to understanding the therapeutic mechanism of D-serine, but rather the role of PV neurons in the underlying cognitive deficits in the MAM model.
6. While MAM animals showed an increased A/N ratio in PV neurons specifically, there is no

evidence that this ratio returned to control levels following D-serine administration. This suggests that D-serine does not directly impact NMDA receptor-mediated current in the restoration of PV excitation, which is in opposition to what the authors argue with respect to the data in Figure 5.

Minor Concerns

1. It would be useful to include control animal electrophysiology recordings in Figure 5 to allow the reader to easily discern if the effects of D-serine in MAM animals restore biophysical properties to control levels. Figures from Figure 1 may be better integrated into the MAM data in Figure 5.
2. The authors claim that this study marks the first of its kind to demonstrate PV-selective NMDA receptor deficits in a non-genetic mouse model for schizophrenia. This claim is misleading because there is no evidence presented in the manuscript that this dysfunction is not found in other interneuron classes. MAM reduces DNA synthesis and targets neurodevelopment, which is likely to have widespread effects across interneuron subtypes. In fact, this has been reported in the rat MAM model. Therefore, conceptually this is not advanced.
3. "Serine" is misspelled in Figure 1P.
4. It appears there is an outlier in the D-serine group in Figure 5F resulting in a significant p value where there otherwise wouldn't be.
5. The introduction of the manuscript suggests that the authors do not anticipate modulation of NMDA receptors to yield much influence upon PV synaptic activity. However, this line of reasoning is hard to follow considering that the authors also suggest throughout the results section that D-serine-mediated changes in PV biophysical properties are likely mediated by NMDA, despite the reasons given in the introduction that this is likely not the case.
6. Some traces were contaminated by 60-Hz noises.

(Huo et al., 2018; Chalkiadaki et al., 2019; Zhang et al., 2021)

Chalkiadaki K, Velli A, Kyriazidis E, Stavroulaki V, Vouvoutsis V, Chatzaki E, Aivaliotis M, Sidiropoulou K (2019) Development of the MAM model of schizophrenia in mice: Sex similarities and differences of hippocampal and prefrontal cortical function. *Neuropharmacology* 144:193-207.
Huo C, Liu X, Zhao J, Zhao T, Huang H, Ye H (2018) Abnormalities in behaviour, histology and prefrontal cortical gene expression profiles relevant to schizophrenia in embryonic day 17 MAM-Exposed C57BL/6 mice. *Neuropharmacology* 140:287-301.
Zhang XQ, Xu L, Ling Y, Hu LB, Huang J, Shen HW (2021) Diminished excitatory synaptic transmission correlates with impaired spatial working memory in neurodevelopmental rodent models of schizophrenia. *Pharmacology, biochemistry, and behavior* 202:173103.

Reviewer #3 (Remarks to the Author):

In this manuscript, Zhang et al. explore the MAM developmental model relevant to schizophrenia in mice, and its interaction D-serine. They find that D-serine differentially impacts PV inhibitory interneurons and pyramidal cells. At the behavioral level they convincingly show that D-serine can rescue behavioral deficits induced in the MAM model. In MAM animals, the authors find that the intrinsic excitability of PV neurons is reduced, as well as NMDAR charge – elegantly linking PV hypo-excitability as well as NMDAR hypofunction to a developmental model of schizophrenia. Furthermore, they observe reduced inhibitory control of pyramidal neurons in MAM animals. The authors show that D-serine can rescue the excitability of PV neurons in the MAM model. Then, to

track down the source of inhibition, the authors use a chemogenetic approach to specifically activate PV neurons. They find that this is sufficient to rescue the behavioral deficits in the MAM animals. Finally, to address changes observed in pyramidal neurons in the MAM model, they use Riluzole to activate SK channels (mimicking the effects of D-serine) and observed behavioral rescue in the MAM model. This manuscript provides exciting and significant insights into possible therapeutic strategies to treat cognitive symptoms of schizophrenia.

This is an impressive body of work that nicely identifies cell-specific dysfunction in a prominent model of schizophrenia. The data presented support the claims made. I have a few, mainly minor, comments that I hope will further strengthen the manuscript:

Major comments:

- It is essential that the authors show the extent of chemogenetic hM3D virus expression with further IHC. This is relevant as the authors mention excluding animals with low expression. Can this be expanded on? Was there a minimal coverage of the ACC that was used as a threshold?
- I appreciate that the authors conduct the rigorous controls within the MAM condition. However, it would be interesting to test if the hM3D manipulation also alters the behavior of control animals. This could help strengthen the narrative that it is PV hypofunction specifically in the MAM condition that needs to be corrected to restore behavioral performance. Alternatively, the possibility that PV excitation in ACC could act to improve PPI, Y-maze and NOR under baseline conditions could be included into the discussion as a caveat.
- Related to the above point, in Figure 7g-n it is important to include a Control/Riluzole group. The authors included all groups when evaluating the effects of D-serine (Figure 2) and it makes the data much easier to interpret.
- It would be very interesting to test if D-serine treatment leads to facilitated NMDAR charge in PV neurons in control animals and in the MAM treated animals. This would help tease out whether the drug effects are through NMDARs or PV cell excitability, or both mechanisms.

Minor comments:

- In Fig 4C the authors show reduced oIPSC amplitude in Pyr cells after optical stimulating PV cells in the MAM condition. The authors note that there is "attenuated inhibitory output". The data is convincing, but it would be useful if the authors could speculate as to whether this is a presynaptic release mechanism, or through failure of Chr2 to excite the PV neurons which are less excitable in the MAM condition.
- Do the authors speculate that the MAM model is causing behavior effects through NMDAR hypofunction in PV neurons? If so, how can this be reconciled with studies showing loss of NR1 from all PV cells has minimal effects on PPI and Y-maze performance? This could be addressed in the discussion with the relevant literature cited.
- More details should be included with respect to the surgery for virus injection. For example, there is no mention of analgesics being given to the animals.
- In the figure legends it would be good to include the number of male or female mice included for each experiment as well as the breakdown of animal number.
- Experimental details are lacking in some sections. For example, it is not clear which virus was used to label PV cells with mCherry for electrophysiology experiments.
- Were the experiments conducted blind to the genotype of animals? This was not mentioned in the manuscript, though it was mentioned in the reporting summary document.
- Was the normality of data tested or assumed? This should be mentioned in the statistics section of the manuscript.

- For the electrophysiology experiments the authors show that D-serine has an acute effect on physiology. In which case, why did the authors choose to use a chronic D-serine regime for the behavioral tests? Is there an acute effect of D-serine in the MAM model? Are there long-term changes in the circuit with D-serine treatment that are required for behavioral rescue? This should be included in the discussion of the manuscript.

- The manuscript contains some grammatical errors that should be fixed.

To Reviewing Editor and Reviewers:

We would like to thank you and the reviewers for your constructive suggestions and insightful critiques, which will help to strengthen the overall impact of our research. We have carefully revised the manuscript in accordance with the points raised by the reviewers. Below is a summary of the additional experiments and analysis we performed.

- 1) We observed whether there was a difference in mAHP amplitude or inhibition rate of mAHP amplitude by apamin between single AP and bursts of APs (i.e., 3-6 APs) protocols. The relevant results were shown in the answer to the corresponding concern.
- 2) We observed how D-serine affects NMDA receptor function, and whether NMDA receptor mediates the effect of D-serine in promoting PV excitability *in vitro*, in both control and MAM groups (Figures 1 and 5).
- 3) We evaluated the impact of D-serine on PV excitability and synapse-driven pyramidal activity (i.e., eAP) on brain slices obtained from the MAM mice that received systemic administration of D-serine (Figure 6).
- 4) We have added whole-slice immunofluorescence images to complement the zoomed-in pictures previously provided in Figure (Figure 7).
- 5) We evaluated the effects of hM3D manipulation and Riluzole on behavioral tests in the control group (Figures S5 and Figure 8).
- 6) To assess the effect of D-serine on spike number in summarized current-firing curves, we conducted two-way ANOVA with repeated measures in both factors (Current Step & Drug) instead of using only one repeated factor (Drug).

Please find attached the revised manuscript, with the changes we have made highlighted in yellow. Additionally, please refer to the following responses to the specific concerns raised by the reviewers.

Reviewer #1:

The paper by Zhang et al. investigates the effects of D-serine in reversing the effects of prenatal MAM treatment. This is a very good and comprehensive paper on the topic. The MAM model is an excellent neurodevelopment model of schizophrenia that allows the detailed investigation of mechanisms, as performed here. However, it cannot be published in the current form. There are methodological/presentation details that need attention and/or elaboration. Furthermore, careful attention should be paid by the authors in the English language. There are several syntactical/grammatical issues and the authors should carefully improve the writing of the paper.

Answer:

Thank you for your positive comment and for bringing the main shortcoming of the initial version to our attention. We have thoroughly revised the paper and the changes are

described in detail in the following sections. Additionally, we engaged a professional help to improve the scientific English writing in the revised version.

Major issues

1. English language: past and present tense are intermixed in the introduction. I think the use of present tense should mostly be used in the introduction.

We apologize for the mistake and have revised the introduction as well as the rest of the manuscript to ensure that the use of tenses is consistent and appropriate. We hope that these revisions have improved the readability of our manuscript.

2. I don't really agree with the results on the mAHP. The mAHP after a single AP is usually not very accurate. The function of SK channels (including mAHP and sAHP) is much better observed and studied after a burst of 3-6APs, usually given following a 50ms step-pulse (see several papers on that such as Matthews EA, Linardakis JM, Disterhoft JF (2009) The fast and slow afterhyperpolarizations are differentially modulated in hippocampal neurons by aging and learning. *The Journal of neuroscience: the official journal of the Society for Neuroscience* 29:4750–4755. Or 1.Liebmann, L. et al. Differential Effects of Corticosterone on the Slow Afterhyperpolarization in the Basolateral Amygdala and CA1 Region: Possible Role of Calcium Channel Subunits. *Journal of Neurophysiology* 99, 958–968 (2008).) On the same point, how reduced mAHP result in reduced excitability? It seems contradictory to me (unless I'm missing something). In most cases and previous studies, reduces the mAHP increased excitability (see references above).

We appreciate your valuable comments concerning the accuracy of mAHP measurements after a single AP and your references to relevant literature. We would like to provide the following rationale for our chosen method of measuring mAHP after a single AP. (1) Studies have showed that mAHP can be generated by a single AP or bursts of APs (Guan et al., 2015; Whitaker et al., 2017; Mi et al., 2019). (2) We observed no difference in the mAHP amplitude between single AP and bursts of APs (i.e., 3-6 APs) protocols, as shown in the figure below. (3) Our results (Fig. 1g) indicated that D-serine could enhance the mAHP amplitude, and the augmented portion was sensitive to apamin, a selective blocker for SK. This observation indicates that mAHP measurements after a single AP are suitable for assessing SK channel function, particularly in the context of D-serine. In the revised version, we provide relevant citations in the Methods section (P21, L 610).

Regarding the 2nd question, the reviewer is correct in stating that a reduced mAHP results in increased excitability. This is because a reduced mAHP would allow for a quicker return to the RMP, which in turn would enable the neuron to fire Aps more frequently. Indeed, the results shown in the Fig 1 indicated that D-serine increased mAHP (Fig 1g) and reduced the pyramidal excitability (Fig 1g & h), which supports the reviewer's statement. However, in describing our findings, we inadvertently made a clerical error by stating that 'we observed that D-serine attenuated the amplitude of medium-duration afterhyperpolarization (mAHP)...'. We appreciate the reviewer's careful attention, and in the revised version, we have corrected this error to read: "we observed that D-serine enhanced the amplitude of medium-duration afterhyperpolarization (mAHP)..." (P4, L97).

3. Details on how the different electrophysiological properties were analyzed should be provided (input resistance, mAHP, etc).

Thank you for your suggestion. We agree that more detailed information on the electrophysiological properties that were analyzed are needed. We have updated the manuscript to include a detailed description of the methods that were used to analyze the data (P21, L603 to 614).

4. In Figure 4, description of the h panel is missing. I assume it indicates sIPSCs following laser activation of PV interneurons. In this case, representative traces from both control and MAM mice should be shown in cases without ontogenetic activation and with ontogenetic activation.

We are grateful for the reviewer's keen attention to detail in Fig. 4. We have included the missing description for the h panel in the revised manuscript. And we would like to clarify that the sIPSCs were indeed recorded without laser activation. In Figure 4f-h, spontaneous IPSCs were recorded from pyramidal neurons adjacent to the mCherry-labeled PV neurons. The observed attenuation of sIPSC frequency indicates the dysfunction of PV neurons in the MAM group.

5. In Figure 5, What is the difference between panels a, b and c in this figure and Figure 1? I guess this is from animals that received D-serine in the water (i.e. systemic D-serine and not in the bath). This should be indicated in the figure legend, and possibly even in the figure.

Thank you for bringing this to our attention. The difference between the panels in Figure 5 and Figure 1 may not have been clear in our initial presentation. We would like to clarify that the data in Figure 5 were indeed sampled from the MAM group, while the data in Figure 1 were sampled from the control group. In both figures, the recorded neurons

were treated with D-serine via bath application. Additionally, new electrophysiological data from mice receiving systemic D-serine administration via their water supply are presented in Fig. 6.

To address the reviewer's concerns, we have made the following revisions: 1) We have updated the figure legends for Figures 1 and 5 to clearly indicate the differences between these two sets of data. 2) For additional clarity, we have also included this information within the figures themselves by adding text annotations.

6. With regards to reporting statistical results, it is important to identify in each experiment the number of neurons and the number of mice used in that experiment. Some of the numbers of the F metrics are extremely high, making me think that they refer to individual action potentials. This is not correct. The authors should average the properties of APs per cell and then perform the statistics on the different cells, always indicating the number of mice used per group as well.

We appreciate the reviewer's attention to the statistical analysis. We would like to clarify that we indeed averaged the properties of APs per cell when performing the statistical analysis for the effect of D-serine on the neuronal excitability. However, it appears that our initial analysis on spike number in summarized current-firing curves inadequate. For example, we assessed the effect of D-serine on pyramidal cell excitability by examining the spiking numbers of 25 cells under 11 distinct current injection steps, both before and after D-serine application. In our initial analysis, we conducted a two-way ANOVA with repeated measures in one factor (Drug: D-serine on/off). However, we should have performed repeated measures in both factors (Current Step & Drug). In the revised version, we have corrected this error. Below is a table showing the degrees of freedom (df) for each source of variation in above case. F distribution has two parameters: degrees of freedom in the numerator (DFn) and the denominator (DFd), which are represented as $F_{(DFn, DFd)}$. DFd could be a high value When it represents the residual *df*.

Source of Variation	df
Current step	10
Drug (D-serine on/off)	1
current x D-serine	10
Subject x current step	240
Subject x Drug	24
Subject	24
Residual	240

We have made the following revisions. 1) In Method section, we have provided a detailed description of our utilization of two-way ANOVA (P23, L653 to 654). 2). We have updated

the statistical results regarding the effect of D-serine on neuronal excitability; however, the main conclusion remains unaltered. 3) We have updated the manuscript to clearly indicate the number of neurons and mice used in each figure legend, as well as summarized in Table S1.

7. To better illustrate the numbers of mice used across experiments, a table with the numbers of mice/cells for each experiment should be provided.

We thank the reviewer's suggestion. A table including the numbers of mice/cells for each experiment has been provided in the supplementary information.

Minor issues

1. D-serine in the water: Were the mice monitored on how much water they were drinking? Was there variability among the mice?

Before the treatment, we monitored water consumption for each mouse. The volume of water consumed by one mouse per day is around 5 mL. And we observed minimal variability in water intake among the mice. Both MAM and control mice were given D-serine (600 mg/L, Sigma-Aldrich, St. Louis, MO, USA) in their drinking water for two weeks, including the period of the behavioral tests. This protocol was based on previous research by Le Douce et al. (2020). It resulted in an approximate daily dose of 100 mg/kg of D-serine for each mouse. Above information has been added to the revised manuscript (P17, L474 to 477).

2. Ln. 477 - NOL should be NOR

We fixed this error in the revision.

3. Ln 500 - when filed with internal. The word solution should be added

We fixed this error in the revision.

4. Were recordings targeted in layer 2, layer 5 or both neurons?

All recorded neurons were specifically targeted in layer V. A detailed description has now been added in the Methods section of the revised manuscript (page 20, paragraph 2).

5. How were the brain slices subjected to immunofluorescence imaged? Besides the zoomed in pictures in figure 6, larger whole slice images should also be provided. This is also true for the optogenetics experiments.

Thank you for the suggestion. We have added whole-slice immunofluorescence images to complement the zoomed-in pictures previously provided in Figure 7 and Suppl. Fig 3 for optogenetics experiments.

6. Figure 1f: is this voltage-clamp or current-clamp. The trace looks like current-clamp but the y-axis is indicated in pA.

We apologize for this typo. Now this error has been fixed.

Reviewer #2

In light of clinical findings supporting NMDA receptor co-agonist D-serine as a novel therapeutic in the treatment of cognitive domains of schizophrenia, Zhang et al. sought to identify the cellular basis for these effects in the MAM mouse model. Given that PV interneuron and NMDA receptor hypofunction in the prefrontal cortex is a common pathological feature of schizophrenia, the authors conducted whole cell electrophysiological recordings in PV and pyramidal neurons to determine the influence of D-serine upon the biophysical properties of these cell types. Additionally, behavioral assays were conducted in the MAM model to assess the efficacy of D-serine in treating cognitive deficits associated with schizophrenia. The authors reported some interesting and novel findings. For example, D-serine was capable of rescuing some behavioral deficits in MAM mice. Bath application of D-serine restored the dysregulated excitability of PV and pyramidal neurons in the ACC of MAM mice, and D-serine-mediated excitability regulation in pyramidal neurons was mediated by SK channel activation. This study thus revealed that treatment with D-serine serves to regulate E/I balance via opposing mechanisms between two relevant cell types. The data presented are well-organized and are generally convincing. However, the impact of these findings is limited by the lack of investigation into the mechanisms by which D-serine facilitates biophysical property changes in PV neurons. There are also some controversial issues with the mice MAM model.

Thank you for the comprehensive overview of our study and for acknowledging the interesting and novel findings we report. We agree with the limitations you mentioned about our findings. We describe in detail below how we addressed these concerns.

Major Concerns

1. First, since the first MAM model was reported by Moore et al. 2006, the general consensus is that MAM administration in rats GD17 produced a relatively stable replication of the pre-clinical symptoms of schizophrenia. In contrast, similar behavioral and physiological changes failed to be reproduced in mice. There were several attempts to develop the MAM model in mice (Chalkiadaki et al, 2019; Hou et al, 2018), including the authors' group (Zhang et al 2021), in recent years. However, considerable variabilities were reported. For example, the dose and timing of MAM administration produced

distinct morphological and physiological changes. There were also different behavioral deficits in males vs. females. These variabilities raise a serious concern that have haunted the field for years. The authors did not address this concern. In particular, the authors did not make it clear whether both male and female offspring were used in the study despite the reported sex differences in morphology, physiology, and behaviors. The authors also did not discuss whether these findings in the mice can be reproduced in the more popular rat MAM model. This concern thus seriously compromises the implication of the study.

We understand the reviewer’s concerns regarding the variability in the MAM mouse model and the issue of sex differences. We've taken these concerns into account in our revised manuscript. Here we would like to explain the changes we have made.

First, it is worth noting that, as reported by Chalkiadaki et al. (2019), the timing of MAM administration (i.e., gestational day 16) is a critical factor for minimizing variability in the mouse model. Our previous publication and preliminary experiments for the current study confirmed this finding. Furthermore, Hou et al. (2018) reported that the C57BL/6 mouse strain was more vulnerable to MAM than rats. Overall, based on our experience, we have consistently observed reproducible behavioral deficits and neuropathophysiological alterations in mice, using our current protocol to generate the MAM model. We have also reviewed available literature using the MAM model in mice and summarized the main findings in the table below (Table 1), which consistently report PPI impairment, indicating impaired sensorimotor gating.

Table1. PPI performance in MAM mice

model	Dose	Time	sex	PPI performance	reference
C57BL/6 mice	26 mg/kg	gestation day (GD) 16	M	reduced PPI	Chalkiadaki et al., 2019
C57/BL6J mice	7.5 mg/kg	GD17	M & F	decrease in PPI with a 69 dB-prepulse intensity	Hou et al., 2018
C57/BL6J mice	10 mg/kg/d*3 days	GD15-17	M	PPI deficits at the post-pubertal period	Takahashi et al., 2019
C57/BL6J mice	26 mg/kg	GD16	M	N/A	Zhang et al., 2021

Regarding the second concern, we want to clarify that both male and female mice were used in our experiments, with nearly equal numbers in each group, as stated in the Methods section of our initial submission and in an additional table (Table S1). We acknowledge the importance of sex differences, but they are beyond the scope of the current study. Furthermore, our study design is not optimized to draw meaningful conclusions on sex differences, which is why we did not conduct *post hoc* sex-based analyses. In the revised version, we have updated the source data file in Excel and Supplementary Information to include the data disaggregated by sex as per the SAGER

guidelines. In the source data file, we have filled the data from male mice in blue and data from females in red, allowing for easy identification.

Furthermore, in the revised version, we have discussed the potential reproducibility of the findings observed in the mice MAM model in rats (P11, paragraph 2).

2. While the authors set out to investigate NMDA hypofunction in PV neurons and the therapeutic potential of D-serine, the experiments did not directly link functional improvements in PV neuron excitability with NMDA function, following D-serine application. This is true of D-serine application in both control and MAM contexts.

Reviewer raises a critical point regarding the direct link between functional improvements in PV excitability and NMDA function following D-serine application. We acknowledge the limitations of our initial submission in establishing such link. Although numerous studies have reported that D-serine can facilitate the activation of NMDA receptors in pyramidal neurons (Shleper et al., 2005; Wolosker, 2006; Neame et al., 2019; Wolosker and Balu, 2020), its precise role in PV neurons remains unclear. To address this concern, we conducted additional experiments to investigate 1) how D-serine affects NMDA receptor function, and 2) whether NMDA receptor mediates the effect of D-serine on PV excitability, in both control and MAM groups.

Our findings revealed that in the MAM group but not in the control group, D-serine had significant impact on NMDA receptor currents in PV neuron and D-APV inhibited the effect of D-serine on PV excitability (Fig. 1 & 5). These results suggest that D-serine and NMDA receptors play a more significant role in PV excitability under prenatal pathogenesis-relevant conditions (i.e., MAM model) rather than under physiologically relevant conditions.

We have provided a more detailed discussion of the potential mechanisms underlying the observed effects of D-serine on PV neuron excitability and their relationship with NMDA receptor function in our study's context (P12 and 13).

3. The therapeutic relevance of D-serine is limited in this study, given that the compound was only applied via bath administration during electrophysiological recordings. Thus, it remains unclear if the restoration in PV excitability would recapitulate systemic administration of D-serine were to be tested. Thus, it is unclear if the behavioral effects of D-serine are due to the restoration of E/I balance in cortical circuits, or some other mechanism.

In response to the reviewer's concern, we evaluated the impact of D-serine on PV excitability and synaptic-driven pyramidal activity (i.e., eAP) on brain slices obtained from

the MAM mice that received systemic administration of D-serine. Our new findings demonstrate that restoration of PV excitability and the absence of burst firing during stimulus trains are observed in MAM mice treated with systemic D-serine, indicating a reestablished E/I balance. These results support our hypothesis that the therapeutic effects of D-serine on cognitive function are mediated, at least in part, by the restoration of E/I balance in cortical circuits. Chronic treatment of D-serine might induce enduring changes in synaptic plasticity, intracellular signaling, and neurocircuitry. Such neuroadaptations contribute to stable improvements in both synaptic and intrinsic inhibitory control, ultimately leading the restoration of E/I balance within cortical circuits, which is crucial for cognitive functioning. We have included the results of these additional experiments (Fig. 6) and relevant discussion in our revised manuscript (P14, L383 to 389).

4. While it is an interesting finding that D-serine administration induces opposing biophysical property changes in pyramidal and PV neurons, the authors did not attempt to further investigate, for example, the precise mechanism by which NMDA receptor co-agonism by D-serine lead to increased excitability in PV neurons. This is problematic, considering the title of the manuscript.

We appreciate the reviewer's suggestion. The findings from our additional experiments demonstrate that in PV neurons sampled from the control group, D-serine does not exert an overall effect on the number of spikes elicited by step current stimulation. Moreover, the firing frequency under baseline conditions or during D-serine incubation is not dependent on the NMDA receptor (Fig. 1). This suggests that NMDA receptor does not contribute to the intrinsic excitability of PV neurons, while it mediates synapse-driven activation of PV neurons and neuronal assemblies (Cornford et al., 2019).

In the other hand, our study primarily aimed to address the key question of how D-serine improves cognitive function in a neurodevelopmental model of schizophrenia by reconstituting inhibitory control in pyramidal neurons. Our findings suggest that D-serine activates SK channels in pyramidal neuron and facilitates PV excitability in the MAM model, leading to the restoration of intrinsic and synaptic inhibitory control. Further investigation into the precise mechanism underlying NMDA receptor-mediated signaling regulation of PV neuron excitability under prenatal pathogenesis-relevant conditions is an important direction of future research that would build on the foundation laid by our study.

In the revised version, we now have added new discussion on these findings (P12, L346 to 352).

5. Chemogenetic experiments shown in the study did not seem relevant to understanding the therapeutic mechanism of D-serine, but rather the role of PV neurons in the underlying cognitive deficits in the MAM model.

We acknowledge that these experiments primarily focus on the role of PV neurons in the underlying cognitive deficits. However, we believe that these experiments are crucial for addressing a key aspect of our study: assessing whether rescuing PV neuron function is sufficient to ameliorate cognitive dysfunction in the MAM model.

As the reviewer pointed out, our results demonstrate that PV dysfunction is a core pathophysiology in the ACC of the MAM group, and that strengthening PV inhibitory control to pyramidal neurons with D-serine could be an effective way to treat impaired cognition. However, the potential involvement of other types of GABAergic interneurons in the MAM model cannot be neglected, and D-serine might improve cognitive performance via benefiting the functional recovery of those neurons. To address this critical question, we employed a chemogenetic strategy to selectively repair PV hypoexcitability in the MAM group.

By demonstrating the sufficiency of rescuing PV neuron function for ameliorating cognitive dysfunction in the MAM model, our chemogenetic experiments provide important insights into the role of PV neurons in the therapeutic effects of D-serine. We believe that these experiments contribute to a more comprehensive understanding of the cellular mechanisms underlying the beneficial effects of D-serine in our neurodevelopmental model of schizophrenia.

We hope that these explanations (P9, L233 to 241) can clarify the rationale behind the chemogenetic experiments and their relevance to understanding the therapeutic mechanism of D-serine in the context of our study.

6. While MAM animals showed an increased A/N ratio in PV neurons specifically, there is no evidence that this ratio returned to control levels following D-serine administration. This suggests that D-serine does not directly impact NMDA receptor-mediated current in the restoration of PV excitation, which is in opposition to what the authors argue with respect to the data in Figure 5.

The reviewer raised an excellent point concerning whether the NMDA receptor mediates the effect of D-serine on PV excitability in MAM mice, which relates to question 2 above. The findings from additional experiments reveal that in the MAM group, NMDAR currents are significantly enhanced by D-serine, which in turn mediates the facilitative effect of D-serine on PV excitability (Fig. 5).

Minor Concerns

1. It would be useful to include control animal electrophysiology recordings in Figure 5 to allow the reader to easily discern if the effects of D-serine in MAM animals restore biophysical properties to control levels. Figures from Figure 1 may be better integrated into the MAM data in Figure 5.

We appreciate your suggestion to improve the clarity of our findings. In the initial manuscript, we separated the data of the control group and the MAM group into Figure 1 and Figure 5, respectively, to avoid overcrowding of data within a single graph, which could make interpretation difficult. However, upon considering your feedback, we have revised the manuscript to better address this concern.

In the updated version, we have included key data, such as the effects of D-serine on the NMDA receptor-mediated currents and neuronal excitability, for both the control and MAM groups in Figure 5. This revised presentation allows readers to more easily compare the key biophysical properties between the two groups and identify whether D-serine restores the properties of the MAM group to those of the control group.

2. The authors claim that this study marks the first of its kind to demonstrate PV-selective NMDA receptor deficits in a non-genetic mouse model for schizophrenia. This claim is misleading because there is no evidence presented in the manuscript that this dysfunction is not found in other interneuron classes. MAM reduces DNA synthesis and targets neurodevelopment, which is likely to have widespread effects across interneuron subtypes. In fact, this has been reported in the rat MAM model. Therefore, conceptually this is not advanced.

We apologize for any confusion our original statement may have caused and acknowledge that the term "PV-selective" may be misleading. Our primary intention was to convey that, relative to pyramidal neurons, NMDA receptor deficits preferably occurred in PV interneurons in the MAM mouse model. We recognize that we did not investigate whether this dysfunction is specific to PV interneurons or also present in other interneuron classes. We have revised our claim in the manuscript to more accurately reflect our findings and the limitations of our study. We now state: 'This study is the first to reveal that, relative to pyramidal neurons, NMDA receptor deficits preferentially occur in PV interneurons in a non-genetic mouse model of schizophrenia.' (P10, L280 to 282)

We understand that MAM-induced neuronal dysfunction may have widespread effects across interneuron subtypes in the MAM mouse model. In this context, it is worth noting that our chemogenetic strategy selectively repaired PV hypoexcitability in the MAM

group, demonstrating the importance of rescuing PV neuron function for ameliorating cognitive dysfunction in this model. We believe that these results provide evidence supporting the distinct role of PV neurons in the MAM mouse model. We hope that our revisions and explanations address the reviewer's concerns and offer a clearer understanding of our findings.

3. "Serine" is misspelled in Figure 1P.

We fixed this error in the revised version.

4. It appears there is an outlier in the D-serine group in Figure 5F resulting in a significant p value where there otherwise wouldn't be.

We appreciate the reviewer's concern regarding the potential outlier in Figure 5F. We used "GraphPad Prism" to identify outliers, but none were detected. Additionally, excluding the "outlier"-like value still resulted in a significant p value (Paired *t*-test: $t_{(21)} = 5.760$, **** $p < 0.0001$; $n = 22$ neurons from 8 MAM group). We hope this addresses the reviewer's concerns.

Identify outliers Summary		A	B
		aCSF	D-serine
		Y	Y
1	Method		
2	ROUT (Q = 0.1%)		
3			
4	Number of points		
5	# Y values analyzed	23	23
6	Outliers	0	0
7			

5. The introduction of the manuscript suggests that the authors do not anticipate modulation of NMDA receptors to yield much influence upon PV synaptic activity. However, this line of reasoning is hard to follow considering that the authors also suggest throughout the results section that D-serine-mediated changes in PV biophysical properties are likely mediated by NMDA, despite the reasons given in the introduction that this is likely not the case.

We apologize for the confusion that may have arisen from our original introduction. We would like to clarify that our intention was to convey the uncertainty surrounding the ability of D-serine to regulate PV function via activating NMDAR, despite the theoretical therapeutic potential of this activation. Clarifying this uncertainty is part of the

motivation behind our work. We have revised the introduction to better deliver this message and avoid any confusion (page 3).

As the reviewer pointed out, our results section does suggest that D-serine-mediated changes in PV biophysical properties are likely mediated by NMDA in the MAM model, despite this not being the case in the control group. This finding was unexpected, given the known limitations of NMDAR's impact on PV synaptic activation.

6. Some traces were contaminated by 60-Hz noises.

Although we did a proper grounding and ran a notch filter, electrical hum was still faintly visible in some traces, especially if the traces were enlarged in Fig 3g. These noises are so small (e.g., 1-2 pA) that they do not affect the detection and quantification of electrical signals from synapses.

(Huo et al., 2018; Chalkiadaki et al., 2019; Zhang et al., 2021)

Chalkiadaki K, Velli A, Kyriazidis E, Stavroulaki V, Vouvoutsis V, Chatzaki E, Aivaliotis M, Sidiropoulou K (2019) Development of the MAM model of schizophrenia in mice: Sex similarities and differences of hippocampal and prefrontal cortical function. *Neuropharmacology* 144:193-207.

Huo C, Liu X, Zhao J, Zhao T, Huang H, Ye H (2018) Abnormalities in behaviour, histology and prefrontal cortical gene expression profiles relevant to schizophrenia in embryonic day 17 MAMExposed C57BL/6 mice. *Neuropharmacology* 140:287-301.

Zhang XQ, Xu L, Ling Y, Hu LB, Huang J, Shen HW (2021) Diminished excitatory synaptic transmission correlates with impaired spatial working memory in neurodevelopmental rodent models of schizophrenia. *Pharmacology, biochemistry, and behavior* 202:173103.

Reviewer #3

In this manuscript, Zhang et al. explore the MAM developmental model relevant to schizophrenia in mice, and its interaction D-serine. They find that D-serine differentially impacts PV inhibitory interneurons and pyramidal cells. At the behavioral level they convincingly show that D-serine can rescue behavioral deficits induced in the MAM model. In MAM animals, the authors find that the intrinsic excitability of PV neurons is reduced, as well as NMDAR charge – elegantly linking PV hypoexcitability as well as NMDAR hypofunction to a developmental model of schizophrenia. Furthermore, they observe reduced inhibitory control of pyramidal neurons in MAM animals. The authors show that D-serine can rescue the excitability of PV neurons in the MAM model. Then, to track down the source of inhibition, the authors use a chemogenetic approach to specifically activate PV neurons. They find that this is sufficient to rescue the behavioral deficits in the MAM animals. Finally, to address

changes observed in pyramidal neurons in the MAM model, they use Riluzole to activate SK channels (mimicking the effects of D-serine) and observed behavioral rescue in the MAM model. This manuscript provides exciting and significant insights into possible therapeutic strategies to treat cognitive symptoms of schizophrenia.

This is an impressive body of work that nicely identifies cell-specific dysfunction in a prominent model of schizophrenia. The data presented support the claims made. I have a few, mainly minor, comments that I hope will further strengthen the manuscript.

We sincerely appreciate your thoughtful and constructive comments on our manuscript. We are glad to learn that you found our work significant and exciting.

Major comments:

1. It is essential that the authors show the extent of chemogenetic hM3D virus expression with further IHC. This is relevant as the authors mention excluding animals with low expression. Can this be expanded on? Was there a minimal coverage of the ACC that was used as a threshold?

Thank you for your valuable suggestion. we have provided new micrographs to better illustrate the extent of hM3D virus expression in the ACC in Figure 7.

Regarding the exclusion criteria for animals, we would like to clarify our original statement from the <Reporting Summary>: 'For viral injection experiments, animals were excluded if the injection was not in the correct region or there was *no expression*'. The reason for "no expression" could be due to the viral vector being injected into the wrong region or an obstruction in the capillary injector. Thus, our exclusion criteria were based on the accuracy of the injection site and presence of virus expression, rather than a specific "threshold" of virus expression. We hope that this clarification addresses your concern.

2. I appreciate that the authors conduct the rigorous controls within the MAM condition. However, it would be interesting to test if the hM3D manipulation also alters the behavior of control animals. This could help strengthen the narrative that it is PV hypofunction specifically in the MAM condition that needs to be corrected to restore behavioral performance. Alternatively, the possibility that PV excitation in ACC could act to improve PPI, Y-maze and NOR under baseline conditions could be included into the discussion as a caveat.

We thank the reviewer for raising this critical point. We have performed behavioral tests in control mice with hM3D manipulation. The results showed no significant differences between hM3D and mCherry groups after CNO injection, indicating that a general

enhancement of PV excitation in the ACC under baseline conditions does not improve cognitive function. These findings support the notion that the behavioral improvements observed in the MAM model are due to the specific correction of PV hypofunction. We have included these new results in the revised manuscript (Fig. S5), along with a discussion of the implications for our study's conclusions (P14, L404 to 407).

3. Related to the above point, in Figure 7g-n it is important to include a Control/Riluzole group. The authors included all groups when evaluating the effects of D-serine (Figure 2) and it makes the data much easier to interpret.

We accept your suggestion. we did conduct additional experiments to evaluate the effects of Riluzole in the control group. Figure 8g-n is updated now.

4. It would be very interesting to test if D-serine treatment leads to facilitated NMDAR charge in PV neurons in control animals and in the MAM treated animals. This would help tease out whether the drug effects are through NMDARs or PV cell excitability, or both mechanisms.

We appreciate your suggestion to test the effects of D-serine treatment on NMDAR charge in PV neurons. The reviewer 2 also raised similar concern on this issue. We have conducted additional experiments using the MAM group, which demonstrates that D-serine facilitates NMDAR charge in PV neurons, and the promoting effect of D-serine on PV excitability can be prevented by D-APV. Integrating the results from the additional experiment, we concluded that NMDARs mediate the restored PV excitability by D-serine in the MAM group (Fig. 5).

As for control animals, the findings from additional experiments demonstrate that in PV neurons, D-serine does not exert an overall effect on the spike numbers elicited by step current stimulation. Moreover, the firing frequency under baseline conditions or during D-serine incubation is independent of the NMDA receptor (Fig. 1). We have incorporated these new findings into our manuscript (P4, L107 to 112).

Minor comments:

1. In Fig 4C the authors show reduced oIPSC amplitude in Pyr cells after optical stimulating PV cells in the MAM condition. The authors note that there is “attenuated inhibitory output”. The data is convincing, but it would be useful if the authors could speculate as to whether this is a presynaptic release mechanism, or through failure of Chr2 to excite the PV neurons which are less excitable in the MAM condition.

The reduced oIPSC amplitude in response to 1st optical stimulus could be due to altered presynaptic release mechanism or altered Chr2 expression (Fig. 4c). The higher ratio of

oIPSC amplitudes evoked by the subsequent stimulus vs. the 1st stimulus can reflect a lower initial probability of neurotransmitter release from PV neurons (Fig. 4e). Thus, we speculate that the 'attenuated inhibitory output' is attributable to a presynaptic release mechanism. Taking these factors into account, we have revised our manuscript to include a more detailed explanation in the Results section (P7, L173 to 175).

2. Do the authors speculate that the MAM model is causing behavior effects through NMDAR hypofunction in PV neurons? If so, how can this be reconciled with studies showing loss of NR1 from all PV cells has minimal effects on PPI and Y-maze performance? This could be addressed in the discussion with the relevant literature cited.

We appreciate this opportunity to discuss the potential discrepancies between our findings and the results of the referenced studies. To address these discrepancies, we have added the following points to our discussion and cited relevant literatures: 1) NMDAR hypofunction in PV neurons may not be the sole mechanism contributing to cognitive impairment in the MAM model. Gestational MAM administration selectively interferes with proliferation and migration of neuronal precursor cells in offspring, leading to a complex and multifaceted prenatal pathogenesis. This suggests that other factors beyond NMDAR hypofunction in PV neurons could contribute to the observed cognitive deficits. 2) While our findings demonstrate that D-serine or the specific correction of PV hypofunction improves cognitive function in the MAM model, this does not preclude the possibility that NMDAR dysfunction in multiple cell types, including PV neurons, other interneurons, and principal neurons, may collectively contribute to the pathophysiology of schizophrenia model. The improvement of cognitive function following the correction of PV hypofunction indicates that this strategy has therapeutic potential, but it does not negate the role of NMDAR dysfunction in other cell types (P11, paragraph 1).

3. More details should be included with respect to the surgery for virus injection. For example, there is no mention of analgesics being given to the animals.

We apologize for the omission of details regarding the surgery in our initial manuscript. We have updated the Materials and Methods section on page 16 with more information about the surgical process, including details about the type of analgesic used, the doses administered.

4. In the figure legends it would be good to include the number of male or female mice included for each experiment as well as the breakdown of animal number.

In response to your critique, we have provided a table include the number of male and female mice used in each experiment in the supplementary information.

5. Experimental details are lacking in some sections. For example, it is not clear which virus was used to label PV cells with mCherry for electrophysiology experiments.

We apologize for any confusion caused by the lack of experimental details in certain sections. The virus was used to label PV cells is AAV2/9-hSyn-DIO-mCherry. In the revised manuscript, we have included this information in the relevant sections (P17, L462).

6. Were the experiments conducted blind to the genotype of animals? This was not mentioned in the manuscript, though it was mentioned in the reporting summary document.

We have updated the Methods section of our revised manuscript to clearly state that (P16, L447 to 449).

7. Was the normality of data tested or assumed? This should be mentioned in the statistics section of the manuscript.

First, we tested the normality of the data to determine whether the mean could be used as a representative value. If the data were found to be normally distributed, we compared means using parametric tests; otherwise, we compared medians using nonparametric methods. Now we have revised the Statistics section of our manuscript to clarify our approach to data normality assessment: an assessment of the normality of data is a prerequisite for choosing parametric or nonparametric testing (P23, L649).

8. For the electrophysiology experiments the authors show that D-serine has an acute effect on physiology. In which case, why did the authors choose to use a chronic D-serine regime for the behavioral tests? Is there an acute effect of D-serine in the MAM model? Are there long-term changes in the circuit with D-serine treatment that are required for behavioral rescue? This should be included in the discussion of the manuscript.

We appreciate the reviewer's suggestion, and we have included a discussion of these points (P14, L383 to 389). It is possible that acute D-serine could also produce beneficial effects on behavior in the MAM model. However, as the reviewer mentioned, chronic treatment of D-serine might induce persistent alteration in synaptic plasticity, intracellular signaling and neurocircuitry. Such neuroadaptation could contribute to stable improvement in synaptic and intrinsic inhibitory control, ultimately resulting in a restoration of E/I balance in cortical circuitry. Importantly, these long-term changes arise from the drug's impact on its primary target, which is the central focus of our study. By emphasizing this connection, we underscore the significance of our research findings in

understanding the therapeutic potential of D-serine for schizophrenia and related disorders.

9. The manuscript contains some grammatical errors that should be fixed.

We apologize for any confusion or difficulty in reading caused by these errors. We have thoroughly reviewed and revised the manuscript to correct any grammatical errors and improve the overall readability. We have also engaged professional help to proofread the manuscript and to improve the scientific English writing in the revised version.

References

Cornford JH, Mercier MS, Leite M, Magloire V, Häusser M, Kullmann DM (2019) Dendritic NMDA receptors in parvalbumin neurons enable strong and stable neuronal assemblies. *Elife* 8.

Guan D, Armstrong WE, Foehring RC (2015) Electrophysiological properties of genetically identified subtypes of layer 5 neocortical pyramidal neurons: Ca²⁺ dependence and differential modulation by norepinephrine. *J Neurophysiol* 113:2014-2032.

Mi Z, Yang J, He Q, Zhang X, Xiao Y, Shu Y (2019) Alterations of Electrophysiological Properties and Ion Channel Expression in Prefrontal Cortex of a Mouse Model of Schizophrenia. *Front Cell Neurosci* 13:554.

Whitaker LR, Warren BL, Venniro M, Harte TC, McPherson KB, Beidel J, Bossert JM, Shaham Y, Bonci A, Hope BT (2017) Bidirectional Modulation of Intrinsic Excitability in Rat Prelimbic Cortex Neuronal Ensembles and Non-Ensembles after Operant Learning. *J Neurosci* 37:8845-8856.

Reviewers' comments:

Reviewer #1 (Remarks to the Author):

The authors have addressed all my concerns.

Reviewer #2 (Remarks to the Author):

Comments to the authors NCOMMS-22-38894

The authors nicely addressed many concerns raised by this reviewer. However, there is still a serious concern about using mouse MAM model for this study. As pointed out in the previous review, the mouse MAM model is not well accepted in the field, and it is very controversial because it failed to reproduce the physiological, morphological, and behavioral changes seen in the well-accepted preclinical rat MAM model. The authors argued that several mouse MAM models reported recently showed consistent PPI impairment, indicating impaired sensorimotor gating in the Rebuttal Table 1. However, it should be noted that although there were several attempts to develop the MAM model in mice (Hou et al, 2018; Takahashi et al, 2019; Chalkiadaki et al, 2019), including the authors' group (Zhang et al, 2021), in recent years, all these papers are not well accepted. These papers were published in relatively low-impact journals, and after being published, none of the papers was highly cited in the field. More importantly, as summarized in the Rebuttal Table 1, these papers reported considerable variability, including using different dosages of MAM (7, 5, 10, 04 26 mg/kg), different gestation days (GD 16 or 17), and different results. The only consistent finding in the mouse MAM model, as listed in the Rebuttal Table 1, is the PPI impairment reported in 3 out of 4 papers (but not by the authors, Zhang et al, 2021). Here is the quote from authors' rebuttal, "as reported by Chalkiadaki et al. (2019), the timing of MAM administration (i.e., gestational day 16) is a critical factor for minimizing variability in the mouse model."

The authors did not consider several other indicators, especially sex difference, reported by Chalkiadaki et al, 2019. For example, Chalkiadaki et al, 2019 reported that only male MAM-16 treated mice showed decreased parvalbumin expression in HPC and PFC. In addition, male, but not female, MAM-16 treated mice exhibited deficits in the delayed alternation task and LTP in layer II PFC synapses. Proteomic analyses of PFC lysates showed significant sex-dependent differences in protein expression regulation. Behaviorally, only male MAM-treated mice have PFC-dependent cognitive deficits. The authors did not efficiently address this critical issue. Instead, they argued that "they are beyond the scope of the current study. Furthermore, our study design is not optimized to draw meaningful conclusions on sex differences, which is why we did not conduct post hoc sex-based analyses."

The fact is that the authors initially ignored the sex difference in their first submission. In this revision, they highlighted the data from male and female mice in the supplemental materials. However, due to the relatively low sample size in many cases (for example, there were only 3 cells from males and 2 cells from the female MAM mouse for Fig 2d and 2e, presumably from 1 or 2 mice each), they mixed the data obtained from both male and female mice. With the female mice not showing reduced PV expression, synaptic plasticity, and behavioral phenotypes (Chalkiadaki et al, 2019), the data presented in this manuscript will likely mislead the readers and the field.

Finally, the authors stated "in the revised version, we have discussed the potential reproducibility of the findings observed in the mice MAM model in rats (P11, paragraph 2)". The discussion is insufficient.

All these concerns thus seriously compromise the implication of the study and will potentially mislead the field.

Reviewer #3 (Remarks to the Author):

I am grateful for the authors detailed and thorough response to my questions and comments. I believe that the revisions have strengthened the main message of the manuscript.

There are many grammatical errors that remain, including but not limited to:

- line 39 where the word "are" should be deleted:

"Abnormalities of parvalbumin-positive (PV) neurons in the frontal cortex are represent a prominent pathology in schizophrenia patients and animal models of schizophrenia."

- Line 246: "recognition memory (interaction of DREADD × CNO by two-way ANOVA, Fig. 67h-n)"

- Line 422:

"Our finding unveiled a novel dual mechanism of D-serine in cognitive improvement via using a neurodevelopmental model of schizophrenia." - the word "via" should be deleted and it should be "findings"

To Reviewers:

We would like to thank the reviewers for your critiques on our previous revision. Although Reviewers #1 and #3 satisfied with the quality of our revisions, we recognized that Reviewer #2 still has reservations about the validity of the mouse MAM model and the alleged neglect of sex differences.

To address the 1st concern from Reviewer #2, we have conducted additional experiments and conduct sex-based analyses to examine potential sex differences in PPI impairment and other key cognitive deficits, and electrophysiology index. And, we have provided a detailed clarification in below to allay the 2nd concern about the model's validity.

Please find the attached point-by-point responses and the revised manuscript with the changes we have made highlighted in yellow.

Reviewer #1:

The authors have addressed all my concerns.

Thank you.

Reviewer #2:

1. The authors nicely addressed many concerns raised by this reviewer. However, there is still a serious concern about using mouse MAM model for this study. As pointed out in the previous review, the mouse MAM model is not well accepted in the field, and it is very controversial because it failed to reproduce the physiological, morphological, and behavioral changes seen in the well-accepted preclinical rat MAM model. The authors argued that several mouse MAM models reported recently showed consistent PPI impairment, indicating impaired sensorimotor gating in the Rebuttal Table 1. However, it should be noted that although there were several attempts to develop the MAM model in mice (Hou et al, 2018; Takahashi et al, 2019; Chalkiadaki et al, 2019), including the authors' group (Zhang et al, 2021), in recent years, all these papers are not well accepted. These papers were published in relatively low-impact journals, and after being published, none of the papers was highly cited in the field.

We appreciate Reviewer #2's concerns about the mouse MAM model's validity and the lower impact of journals that published earlier related studies. We would like to emphasize that these independent studies (Hou et al, 2018; Takahashi et al, 2019; Chalkiadaki et al, 2019), which corroborated findings of the mouse MAM model demonstrating consistent PPI impairment - a gold standard for schizophrenia animal models - were published in '*Neuropharmacology*', a respected journal in the field of

neuropsychopharmacology. The quality and reliability of an individual study, we contend, should also be assessed based on its scientific rigor and merit.

According to publisher ELSEVIER, these papers have earned 13, 19, and 24 citations, suggesting their recognition and acceptance in the field. *We hold the view that science is an iterative and cumulative process, with advances often built upon the collective work of the community.* The relatively lower citation rates of these studies may reflect the nascent status of this particular research field - that is, the study of schizophrenia in mouse models via non-genetic and prenatal pathogenesis-relevant conditions.

2. More importantly, as summarized in the Rebuttal Table 1, these papers reported considerable variability, including using different dosages of MAM (7, 5, 10, or 26 mg/kg), different gestation days (GD 16 or 17), and different results. The only consistent finding in the mouse MAM model, as listed in the Rebuttal Table 1, is the PPI impairment reported in 3 out of 4 papers (but not by the authors, Zhang et al, 2021). Here is the quote from authors' rebuttal, "as reported by Chalkiadaki et al. (2019), the timing of MAM administration (i.e., gestational day 16) is a critical factor for minimizing variability in the mouse model."

The reason why we chose dose (26 mg/kg) of MAM is not only to refer to the paper of Chalkiadaki et al. (2019), but most of the studies using the rat MAM model also used a dose of 20-25 mg/kg. More importantly, in both our previous and current study, we have further validated the claim that prenatal MAM (GD 16; 26 mg/kg) impairs PPI. Given the model's established reliability, we did not include the PPI data in our previous study (published in '*Pharmacol Biochem Behav*', 2021). However, in the present study, we have incorporated both male and female mice in almost equal numbers across all experiment groups. We have once again found that the MAM group displays impaired PPI. Hence, we stand firm in our assertion that the MAM model are valid.

3. The authors did not consider several other indicators, especially sex difference, reported by Chalkiadaki et al, 2019. For example, Chalkiadaki et al, 2019 reported that only male MAM-16 treated mice showed decreased parvalbumin expression in HPC and PFC. In addition, male, but not female, MAM-16 treated mice exhibited deficits in the delayed alternation task and LTP in layer II PFC synapses. Proteomic analyses of PFC lysates showed significant sex-dependent differences in protein expression regulation. Behaviorally, only male MAM-treated mice have PFC-dependent cognitive deficits. The authors did not efficiently address this critical issue. Instead, they argued that "they are beyond the scope of the current study. Furthermore, our study design is not optimized to draw meaningful conclusions on sex differences, which is why we did not conduct post hoc sex-based analyses."

In terms of the concern regarding sex differences, we have complied with the 'The Sex and Gender Equity in Research (SAGER) guidelines' established by the Springer Nature. As the guidelines caution against post hoc sex-based analyses when the study design is not optimized for meaningful conclusions, we refrained from such analysis in initial version and previous revision. Nonetheless, we appreciate the reviewer's perspective on this matter. In this revised version, we expanded the sample size to further investigate the potential sex differences in cognitive impairment and baseline neurophysiological abnormalities resulting from prenatal MAM exposure. Our sex-based analysis (details provided in below table) includes at least six animals per sex group for behavioral experiments and five for electrophysiological experiments. We found no significant sex-dependent variations in cognitive dysfunction or abnormal neurophysiological indices attributable to prenatal MAM (Table S1). Therefore, we did not extend our investigation to include potential sex differences in the therapeutic effects of D-serine or other drugs.

Figure	Behavioral Experiment or Electrophysiological Index	Number of cells	Number of mice
Fig. 2b-c	PPI	/	Control: 12 (6M, 6F); MAM: 13 (7M, 6F).
Fig. 2d	open field	/	Control: 13 (7M, 6F); MAM: 13 (7M, 6F).
Fig. 2e-f	Y maze	/	Control: 13 (7M, 6F); MAM: 16 (8M, 8F).
Fig. 2g-h	novel location/ object recognition	/	Control: 13 (7M, 6F); MAM: 13 (7M, 6F).
Fig. 3b-f	PV neuron: 1. spike number under 200 pA of depolarizing current 2. Rheobase 3. RMP 4. Input resistance	Control: 21; MAM: 23	Control: 10 (5M, 5F); MAM: 10 (5M, 5F)
Fig. 3i-j	PV neuron: NMADR-EPSC charges, A/N ratios	Control: 20; MAM: 20	Control: 10 (5M, 5F); MAM: 10 (5M, 5F)
Fig. S2b	Pyramidal neuron: spike number under 300 pA of depolarizing current	Control: 29; MAM: 31	Control: 10 (5M, 5F); MAM: 10 (5M, 5F)
Fig. S2d & e	Pyramidal neuron: NMADR-EPSC charges, A/N ratios	Control: 19; MAM: 18	Control: 10 (5M, 5F); MAM: 10 (5M, 5F)

- The fact is that the authors initially ignored the sex difference in their first submission. In this revision, they highlighted the data from male and female mice in the supplemental materials. However, due to the relatively low sample size in many cases (for example, there were only 3 cells from males and 2 cells from the female MAM mouse for Fig 2d and 2e, presumably from 1 or 2 mice each), they mixed the data obtained from both male and female mice. With the female mice not showing reduced PV expression, synaptic

plasticity, and behavioral phenotypes (Chalkiadaki et al, 2019), the data presented in this manuscript will likely mislead the readers and the field.

In previous revision, we have indicated the number of neurons and mice used in each figure legend, as well as summarized in Supplementary Table 1. For instances where our data presented low sample sizes, including Supplementary Fig 2d and 2e you mentioned, we have increased the number of neurons and animals in both sexes to investigate the potential sex differences in current revision (Supplementary Table 2).

5. Finally, the authors stated “in the revised version, we have discussed the potential reproducibility of the findings observed in the mice MAM model in rats (P11, paragraph 2)”. The discussion is insufficient.

In current revision, we have included additional discussion regarding the sex differences caused by prenatal MAM.

6. All these concerns thus seriously compromise the implication of the study and will potentially mislead the field.

Now, we have expanded the sample size to conduct sex-based analyses to examine potential sex differences in PPI impairment and other key cognitive deficits, and electrophysiology indices. And, we have provided a detailed clarification about the model's validity. We hope these efforts be able to address reviewer #2's concerns.

Reviewer #3:

1. I am grateful for the authors detailed and thorough response to my questions and comments. I believe that the revisions have strengthened the main message of the manuscript.

Thank you.

2. There are many grammatical errors that remain, including but not limited to:
 - line 39 where the word "are" should be deleted:
"Abnormalities of parvalbumin-positive (PV) neurons in the frontal cortex are represent a prominent pathology in schizophrenia patients and animal models of schizophrenia."
 - Line 246: "recognition memory (interaction of DREADD × CNO by two-way ANOVA, Fig. 67h-n)"
 - Line 422:

"Our finding unveiled a novel dual mechanism of D-serine in cognitive improvement via using a neurodevelopmental model of schizophrenia." - the word "via" should be deleted and it should be "findings"

We appreciate your close reading of our manuscript. We have thoroughly checked the manuscript again and corrected several grammatical errors.

3. Beyond typographical edits I have no further comments or concerns.

Thank you.

REVIEWER COMMENTS

Reviewer #4 (Remarks to the Author):

This is an interesting report. The authors aimed to elucidate the cellular mechanism underlying the therapeutic effect of D-serine as potential treatment for cognitive impairment related to schizophrenia. To do this, the authors first tested the effects of D-serine on pyramidal neurons in brain slices from control animals and found that D-serine decreased ACC pyramidal neuron excitability in a SK-channel dependent manner (Figure 1). Next, the authors utilized a MAM mouse model related to schizophrenia, and demonstrated numerous types of cognitive deficits that can be ameliorated by chronic D-serine treatment (Figure 2). In the subsequent experiments, the authors continued to show that: a) MAM mice had reduced ACC pyramidal neuron excitability (Figure 3), b) MAM mice had reduced local inhibitory control from PV cells (Figure 4), and c) in ACC slices from MAM mice, D-serine can increase PV neuron excitability while reducing pyramidal neuron excitability (Figure 5). Based on such behavioral and in vitro evidence, the researchers next demonstrated the therapeutic efficacy/potential of D-serine systemic treatment (Figure 6), in vivo ACC PV neuron DREADDs activation (Figure 7), and a SK-channel activator (riluzole; Figure 8). Collectively, the authors concluded that there is a "dual mechanism" of D-serine in cognitive improvement, in the sense that "D-serine reconstituted synaptic and intrinsic inhibitory control of cingulate pyramidal neurons via facilitating PV excitability and activating pyramidal SK channels, respectively."

Although each experiment appeared sufficiently powered and well carried out, this study as a whole does not seem to support the above-mentioned conclusion. This is largely due to the alternative hypotheses are not sufficiently considered or discussed. There are also several occasions of missing critical citations, important experimental details, or important control groups. Specific comments are below:

Major issues:

1. I am partially in line with previous reviewer's comments on using the MAM mouse model. In humans, direct MAM exposure (or DNA alkylating agents) is not recognized as a direct risk for schizophrenia. In animal research, like other environmental factors (e.g., maternal immune activation), prenatal MAM exposure only partially recapitulates schizophrenia-related changes, with pathophysiological changes in dopamine system being the most robust finding. Furthermore, as the authors mentioned in the previous rebuttal letter, MAM's effects depend highly on dose, gestation time, and rearing environment. Thus, at the current knowledge state, whether MAM mouse model has sufficient correspondence to human schizophrenia-related pathology and pathophysiology, or cognition, is unclear. Therefore, in general more precise language should be used in this manuscript, and the interpretation of the results should be more objective. For example, the use of "MAM model 'of' schizophrenia" throughout this manuscript is misleading, which should be replaced by "MAM model 'for' schizophrenia" or "... to study schizophrenia". The tested MAM exposure paradigm (that is, 26mg/kg at GD16) clearly induced cognitive deficits in adults and PV-related pathology, which provides a useful tool to test the mechanisms of D-serine in cognitive improvement relevant to schizophrenia but also to many other disorders. The authors should discuss their results deemphasizing schizophrenia.

2. This study used a daily dose of D-serine at 100mg/kg [Line 133 and 134]. The authors cited only one article [Ref 26] and stated their paradigm is proven to increase brain D-serine level. The rationale here is unclear, because the cited article used a much longer (35-day) paradigm at a much lower daily dose, 58 mg/kg. As a result, these two studies are not comparable. Please provide clear rationale for selecting D-serine doses. If no previous study supports the paradigm here increases brain D-serine levels, HPLC or similar assays must be done. This would be the foundation for Figure 2 and 6, and without it several alternative interpretations could be true.

3. What are the purposes of a separate control experiments in Figure 1? Why are these control animals separated from Figure 5g-l, where the authors directly tested both control and MAM mice? Is this a separate set of control animals from those in Figure 5g? Are there inter-experiment differences between the control groups in Figure 1 vs. Figure 5.

4. Related to previous point, please make clear indication in Fig 5.g-j that these panels are from MAM mice.

5. Why are control data only showing in Fig. 5k but not in other panels?

6. The conclusion that D-serine restored ACC pyramidal neuron intrinsic excitability "via" SK channel in MAM mice [Line 254, Line 433] is overgeneralized, and current data does not necessarily support this causal relationship. While apamin was shown to block D-serine's action in many experiments, pharmacological properties of apamin were not sufficiently discussed. Thus, apamin results in slices may only suggest that some effects of D-serine on pyramidal neurons in MAM mice can be blocked by one particular SK channel antagonist. Additional experiments with other SK channel antagonists are required for mechanism of action.

7. Related to the previous point, is apamin at the chosen dose (2uM) selectively block SK channel? The cited paper (Ref 56) utilized at different concentration of 300nM, so what is the rationale of choosing a lower concentration here?

8. Figure 6 was not done as comprehensively as previous recording experiments. For example, only changes in interneuron excitability are reported, but what about pyramidal neurons? In addition, whether data in Fig. 6d indicates a clear "restoration" effect is questionable, because MAM+/D-serine- and MAM+/D-serine+ group did not seem to significantly differ. This could mean only a partial rescue. Please discuss Figure 6 in a more comprehensive manner. Similarly, there is not any saline control groups in Fig 6f, so is the observed effects in MAM+, D-serine+ group truly a restoration of E/I balance? What is the rationale of only using the MAM mice?

9. Some other experiments and experimental details appeared missing:

a. In Figure 5, why are apamin response only shown for mAHP analysis (Fig 5k) but not other endpoints? Such as 250pA current step response?

b. Figure 4, the reduced oIPSC in MAM mice can be from multiple sources. For example, is there reduced overall PV-cell density? There are documented cortical shrinkage and change in cell densities of MAM models, would that affect the sampling in this experiment? Please provide micrographs (DIC images) showing recording site and morphologies of recorded neurons.

c. In all recording experiments, are different subtypes (i.e., intratelencephalic vs. pyramidal tract) of deep layer pyramidal neurons considered? These subtypes of pyramidal neurons tend to exhibit differential inhibitive connections (e.g., PMID: 24361076, PMID: 29798890). Is this controlled in the sampling process in the present study?

Minor issues:

Several key references appeared missing, which limits the interpretation of the current data. Examples are shown below.

1. Line 97-98, "Generally, mAHP is largely modulated by small-conductance Ca²⁺-activated K⁺ (SK) channels and limits firing frequency". Citations are missing.

2. Line 335-339, important citations appeared missing.

3. Line 256-257, missing references for riluzole as a SK channel activator.

4. Line 268-271, missing references.

Mechanism of MAM and its use to model schizophrenia-related changes should be discussed in more detail.

5. In Introduction, Line 76 and 77 state that "to mimic prenatal pathogenesis, pregnant mice were injected with DNA-alkylating agent methylazoxymethanol acetate (MAM), and their offspring were utilized as a neurodevelopmental model of schizophrenia". "Prenatal pathogenesis" is a vague term, and the authors should make clear explanation how MAM exposure is related to schizophrenia prenatal pathogenesis.

6. Similarly, Line 129 and 130 write "[g]estational MAM administration is utilized to model prenatal neurodevelopmental disorders, one hypothesis for schizophrenia." Authors should be clear on what hypothesis is referred here and provide clear citations.

For reproducibility, please address below issues:

7. For Figure 1, please include a representative figure showing the morphology of the recorded neurons, their laminar location, border of ACC, and location of stimulating electrode and recording pipette.

8. For optogenetics, please clarify whether a laser or a LED was used as the light source. What was the light power? What was the filter setting?

9. After last revision, I appreciate that the authors now provide total animal and total cell number

information for each figure, but on average how many cells are sampled from each animal? Was there effort to limit pseudo-replication? How many cells can come from a single animal? From the supplementary dataset, it appears that only 1-3 cells were from each animal, but this needs to be clearly stated for rigor and reproducibility.

There is still room for improvement for grammar and clarity, for example:

10. Line 367: "dose" should be "does"

11. Line 410: "couldn't" should be "could not"

To Reviewing Editor and Reviewer:

Thank you for your detailed feedback and constructive suggestions. We have carefully reviewed your comments and revised our manuscript accordingly. Below is our point-by-point response.

Reviewer #4

Overall comment:

This is an interesting report. The authors aimed to elucidate the cellular mechanism underlying the therapeutic effect of D-serine as potential treatment for cognitive impairment related to schizophrenia. To do this, the authors first tested the effects of D-serine on pyramidal neurons in brain slices from control animals and found that D-serine decreased ACC pyramidal neuron excitability in a SK-channel dependent manner (Figure 1). Next, the authors utilized a MAM mouse model related to schizophrenia, and demonstrated numerous types of cognitive deficits that can be ameliorated by chronic D-serine treatment (Figure 2). In the subsequent experiments, the authors continued to show that: a) MAM mice had reduced ACC pyramidal neuron excitability (Figure 3), b) MAM mice had reduced local inhibitory control from PV cells (Figure 4), and c) in ACC slices from MAM mice, D-serine can increase PV neuron excitability while reducing pyramidal neuron excitability (Figure 5). Based on such behavioral and in vitro evidence, the researchers next demonstrated the therapeutic efficacy/potential of D-serine systemic treatment (Figure 6), in vivo ACC PV neuron DREADDs activation (Figure 7), and a SK-channel activator (riluzole; Figure 8). Collectively, the authors concluded that there is a “dual mechanism” of D-serine in cognitive improvement, in the sense that “D-serine reconstituted synaptic and intrinsic inhibitory control of cingulate pyramidal neurons via facilitating PV excitability and activating pyramidal SK channels, respectively.”

Although each experiment appeared sufficiently powered and well carried out, this study as a whole does not seem to support the above-mentioned conclusion. This is largely due to the alternative hypotheses are not sufficiently considered or discussed. There are also several occasions of missing critical citations, important experimental details, or important control groups.

We are gratified that you find our study interesting. We understand your concerns regarding the conclusions drawn, as well as the consideration of alternative hypotheses and missing details. We attempt to address each point head-on below.

Major issues:

1. I am partially in line with previous reviewer’s comments on using the MAM mouse model. In humans, direct MAM exposure (or DNA alkylating agents) is not recognized as a direct risk for schizophrenia. In animal research, like other environmental factors (e.g., maternal

immune activation), prenatal MAM exposure only partially recapitulates schizophrenia-related changes, with pathophysiological changes in dopamine system being the most robust finding. Furthermore, as the authors mentioned in the previous rebuttal letter, MAM's effects depend highly on dose, gestation time, and rearing environment. Thus, at the current knowledge state, whether MAM mouse model has sufficient correspondence to human schizophrenia-related pathology and pathophysiology, or cognition, is unclear. Therefore, in general more precise language should be used in this manuscript, and the interpretation of the results should be more objective. For example, the use of "MAM model 'of' schizophrenia" throughout this manuscript is misleading, which should be replaced by "MAM model 'for' schizophrenia" or "... to study schizophrenia". The tested MAM exposure paradigm (that is, 26mg/kg at GD16) clearly induced cognitive deficits in adults and PV-related pathology, which provides a useful tool to test the mechanisms of D-serine in cognitive improvement relevant to schizophrenia but also to many other disorders. The authors should discuss their results deemphasizing schizophrenia.

Your suggestion on the interpretation of our experimental model is well-taken. We agree that the MAM mouse model is not a model "of" schizophrenia, but rather serves as a tool to study potential aspects of schizophrenia given its capability to simulate certain pathological and pathophysiological changes. We have now revised our manuscript to replace the terminology "model of schizophrenia" with "model for schizophrenia" throughout.

Regarding your suggestion to discuss the results while de-emphasizing schizophrenia, we adjusted our conclusion accordingly (page11, line 296-299; page16, line 461-464). For example, the revised conclusion will now read: "These findings suggest that restoration of inhibitory control over cortical microcircuit may be an efficacious strategy for the treatment of cognitive impairment seen in conditions where NMDAR deficit-induced PV neuron dysfunction may be a pathological feature."

2. This study used a daily dose of D-serine at 100mg/kg [Line 133 and 134]. The authors cited only one article [Ref 26] and stated their paradigm is proven to increase brain D-serine level. The rationale here is unclear, because the cited article used a much longer (35-day) paradigm at a much lower daily dose, 58 mg/kg. As a result, these two studies are not comparable. Please provide clear rationale for selecting D-serine doses. If no previous study supports the paradigm here increases brain D-serine levels, HPLC or similar assays must be done. This would be the foundation for Figure 2 and 6, and without it several alternative interpretations could be true.

As you pointed out, we indeed cited a separate article (Ref 26 old) in the Results section wherein a different dosage regimen of D-serine, 58mg/kg for 35 days, was implemented. This reference was provided to highlight the therapeutic efficacy of D-serine in improving cognitive function.

However, the rationale for selecting the dosage of D-serine in our own experimental setting comes from a different study, mentioned in the Methods section (page 18 line 509-511, PMID: 32130882). This study confirmed, using HPLC, that a dose of D-serine at 100mg/kg significantly increased D-serine levels in the brain. We admit to the possibility of creating confusion by referring to two different studies with varying doses of D-serine. Now we have replaced citation to avoid any misunderstanding (page 5, line 139; Ref 28 new).

3. (1) What are the purposes of a separate control experiments in Figure 1? (2) Why are these control animals separated from Figure 5g-l? where the authors directly tested both control and MAM mice? (3) Is this a separate set of control animals from those in Figure 5g? Are there inter-experiment differences between the control groups in Figure 1 vs. Figure 5.

We hope that the following clarifications adequately address your concerns.

- (1) In Figure 1, we primarily sought to evaluate the baseline status of D-serine's impact on the excitability of cingulate pyramidal and PV neurons within the control group, establishing a fundamental understanding of D-serine's physiological role in regulating neuronal excitability and SK channel activity. Therefore, we allocated a subset of the control group specifically for this initial investigation.
- (2) In Figure 5g-l, our aim was to elucidate the effect of D-serine on pyramidal neurons from the MAM model of schizophrenia. Direct comparison of baselines of key biophysical properties in pyramidal neurons, including neuronal excitability and NMDA receptor-mediated currents, between the control and MAM groups was shown in Supplementary Figure 2, and no significant differences were detected. We included key data, such as the effects of D-serine on neuronal excitability, for both the control and MAM groups in Figure 5k. D-serine exhibited a similar impact on spike number elicited by 250 pA of depolarizing current across both groups (Fig. 5k).
- (3) Indeed, they are separate sets of mice allocated specifically for each control group. No inter-experiment differences were observed between the control groups in Figure 1 vs. Figure 5. Detailed data supporting this are provided in Source data in the Supplementary materials.

4. Related to previous point, please make clear indication in Fig 5.g-j that these panels are from MAM mice.

We apologize for this oversight that caused confusion. We have added clear indications in Figure 5.g-j clarifying those panels are from MAM mice.

5. Why are control data only showing in Fig. 5k but not in other panels?

Thank you for your question. Comparisons of key biophysical properties at baseline such as neuronal excitability and NMDA receptor-mediated currents in pyramidal neurons between the control and MAM groups have been presented in Supplementary Figure 2. We did not

compare the baseline mAHP (Fig. 5j) in pyramidal neurons between the control and MAM groups due to their similar excitability. Similarly, the baseline eAP number was not compared (Fig. 5l) between the control and MAM groups because the stimulus strength was adjusted to elicit 4-6 firings per stimulus train (10 pulses at 20 Hz, see the Method), making the baseline eAP number a relatively stable value across groups. Notably, a minor fraction of the pyramidal neurons in the MAM group displayed a probability of burst firing under the eAP protocol (Fig. 4j). We hope this clarification addresses your concerns.

6. The conclusion that D-serine restored ACC pyramidal neuron intrinsic excitability “via” SK channel in MAM mice [Line 254, Line 433] is overgeneralized, and current data does not necessarily support this causal relationship. While apamin was shown to block D-serine’s action in many experiments, pharmacological properties of apamin were not sufficiently discussed. Thus, apamin results in slices may only suggest that some effects of D-serine on pyramidal neurons in MAM mice can be blocked by one particular SK channel antagonist. Additional experiments with other SK channel antagonists are required for mechanism of action.

We acknowledge that our data demonstrated a correlation rather than a direct causal relationship between D-serine's effect on the intrinsic excitability of pyramidal neurons and SK channel activation. We have employed apamin, a well-characterized, potent, and highly selective inhibitor of SK channels, to demonstrate this. Indeed, apamin has been recognized as a prototypical SK channel inhibitor, capable of blockage at nanomolar or even subnanomolar concentrations, with no significant effects on other molecular targets, which we did not explicitly mention in the original manuscript (more discussion now; page 15, line 423-427) ¹.

We appreciate the recommendation for further study involving additional SK channel antagonists. However, we feel that our current data already provide compelling evidence for the involvement of SK channels in the mechanism of D-serine's effects in MAM mice, based on the established selectivity of apamin for these channels. Nonetheless, we agree that it may still have unforeseen actions on other molecular targets in our experimental setting. Therefore, we completely agree that this does not categorically exclude the involvement of other components. Considering this feedback, we modified the wording in the relevant text of our manuscript to fairly represent our findings and incorporate the outlined limitations. We change the phrasing from "via SK channels" to "potentially mediated by SK channels" that clearly conveys the correlative (page 11, line 294), rather than strictly causal relationship, while implying the likelihood of other components being involved (page 15, line 427-430).

This change does not detract from our primary conclusions nor diminish the importance of our findings; instead, it honestly presents the reality that our understanding of the mechanisms underlying the impact of D-serine on intrinsic excitability is still evolving. We

believe this modification resolves the issues noted by the reviewer without necessitating additional experimental work at this point.

7. Related to the previous point, is apamin at the chosen dose (2 μ M) selectively block SK channel? The cited paper (Ref 56) utilized at different concentration of 300nM, so what is the rationale of choosing a lower concentration here?

There seems to be a misunderstanding concerning the concentration of apamin used in our study. As outlined in the 'Methods' section of our original manuscript, we used apamin at a concentration of 0.2 μ M, not 2 μ M. Our decision to use apamin at this specific concentration is supported by several studies²⁻⁴, which have effectively used this similar concentration for their investigations.

The paper we cited (Ref 56) in the discussion indeed used a higher concentration of apamin (0.3 μ M). This reference was chosen because we followed their documented protocol for measuring the mAHP amplitude. We regret any confusion caused by our citation choice. To dispel any further misunderstanding, we have now removed this citation and referenced other sources that use a similar concentration of apamin (page 23, the 1st line).

8. (1) Figure 6 was not done as comprehensively as previous recording experiments. For example, only changes in interneuron excitability are reported, but what about pyramidal neurons? (2) In addition, whether data in Fig. 6d indicates a clear “restoration” effect is questionable, because MAM+/D-serine- and MAM+/D-serine+ group did not seem to significantly differ. This could mean only a partial rescue. Please discuss Figure 6 in a more comprehensive manner. (3) Similarly, there is not any saline control groups in Fig 6f, so is the observed effects in MAM+, D-serine+ group truly a restoration of E/I balance? What is the rationale of only using the MAM mice?

(1) We did not examine whether systemic D-serine administration could restore pyramidal excitability in the MAM group (Fig. 6) because no significant differences in pyramidal excitability were observed between the control and MAM groups (Supplementary Figure 2).

(2) Regarding Fig. 6d, we understand your point about whether the data indicates a clear “restoration”. The *post hoc* test showed that firing frequency in the MAM+/D-serine+ group was statistically comparable to that in the control group, which suggests a rebound effect. We recognize that there was no significant difference between the MAM+/D-serine- and MAM+/D-serine+ groups, suggesting that this could be a partial rescue only. We now provided an additional discussion surrounding this result (page 8-9, line 225-231).

(3) We appreciate the reviewer's attention to the absence of a saline control in Fig. 6f. Our experimental setup utilized an eAP stimulation protocol calibrated to elicit 40-60% of the maximum eAP response, inducing 4-6 firings per stimulus train comprised of 10 pulses at

20 Hz. This protocol did not elicit burst firing in the control group (Fig. 4j). Given this, including a saline control group in Fig. 6f would not have significantly contributed additional data towards our primary aim of evaluating the rectifying effects of D-serine on the increased probability of burst firing under eAP protocol in MAM group.

9. Some other experiments and experimental details appeared missing:

a. In Figure 5, why are apamin response only shown for mAHP analysis (Fig 5k) but not other endpoints? Such as 250pA current step response?

We have included data illustrating the effect of apamin on the 250pA current step response in Fig. 5i in the last version. We hope this clarification addresses your question.

b. Figure 4, the reduced oIPSC in MAM mice can be from multiple sources. For example, is there reduced overall PV-cell density? There are documented cortical shrinkage and change in cell densities of MAM models, would that affect the sampling in this experiment? Please provide micrographs (DIC images) showing recording site and morphologies of recorded neurons.

We agree with your opinion. The reduced oIPSC in MAM mice reflect less inhibitory input from PV neurons. Our data already indicate a functional alteration - specifically, hypoexcitability of PV neurons in the MAM group - which can partially explain reduced inhibitory output (oIPSCs). However, multiple potential sources might contribute this reduction, including PV neuron dysfunction, cortical shrinkage, less PV neuron density^{5,6}. Now we have provided additional discussion on this issue (page 14, line 398-400).

In the revised manuscript, we have added a new representative image to Figure 1, which shows both the Infrared DIC image coupled with fluorescence of the recorded neurons and recording site.

c. In all recording experiments, are different subtypes (i.e., intratelencephalic vs. pyramidal tract) of deep layer pyramidal neurons considered? These subtypes of pyramidal neurons tend to exhibit differential inhibitive connections (e.g., PMID: 24361076, PMID: 29798890). Is this controlled in the sampling process in the present study?

The reviewer raised an excellent point. We did not sort out the subtypes of pyramidal neurons in our recording experiments. Actually, not all regions of the cerebral cortex contain pyramidal neurons that project to subcortical areas via the pyramidal tract, specifically the spinal cord. In the anterior cingulate cortex (ACC), which is implicated more in cognitive and emotional processing, the majority of the projection neurons are Intratelencephalic and do not send axons to the spinal cord. Pyramidal tract neurons are primarily found in the primary motor cortex and other related areas producing motor signals. Therefore, it would not be

accurate to differentiate to these neurons as "intratelencephalic vs. pyramidal tract". However, we acknowledge the substantial heterogeneity among Layer-5 pyramidal neurons in the prefrontal cortex. In the revision, our findings have been discussed with the caveat that different subtypes of pyramidal neurons might have been mixed in the sampled population, adding a potential source of variability to the measurements (page11, line 296-305).

Minor issues:

Several key references appeared missing, which limits the interpretation of the current data. Examples are shown below.

1. Line 97-98, "Generally, mAHP is largely modulated by small-conductance Ca²⁺-activated K⁺ (SK) channels and limits firing frequency". Citations are missing.

We have added relevant references to support this statement (PMID: 14500775).

2. Line 335-339, important citations appeared missing.

We have added the appropriate citations to these sections (PMID: 12015199, PMID: 20194128).

3. Line 256-257, missing references for riluzole as a SK channel activator.

We have added relevant references to support this statement (PMID: 12163105).

4. Line 268-271, missing references.

We have added the appropriate citations to these sections (PMID: 25540902 , PMID: 34658976)

Mechanism of MAM and its use to model schizophrenia-related changes should be discussed in more detail.

In the introduction of revised manuscript, we have expanded explanation on the mechanism of MAM (page 3, line 77-80). MAM is a DNA-alkylating agent that interfere with DNA synthesis in proliferating cells. MAM is administered to pregnant rodents during a specific gestational window to mimics developmental disruption. Thus, it is often employed to generate animal models of neurodevelopmental disorders, such as schizophrenia ⁷.

5. In Introduction, Line 76 and 77 state that "to mimic prenatal pathogenesis, pregnant mice were injected with DNA-alkylating agent methylazoxymethanol acetate (MAM), and their offspring were utilized as a neurodevelopmental model of schizophrenia". "Prenatal pathogenesis" is a vague term, and the authors should make clear explanation how MAM exposure is related to schizophrenia prenatal pathogenesis.

In this study, "prenatal pathogenesis" refers to the detrimental events or disruptions occurring during the prenatal period that can lead to neurodevelopmental disorders like schizophrenia. We have expanded on this concept in above response to minor issues 4, and

the detailed explanation can be found in the Introduction in the revised manuscript (page 3, line 77-80).

6. Similarly, Line 129 and 130 write “[g]estational MAM administration is utilized to model prenatal neurodevelopmental disorders, one hypothesis for schizophrenia.” Authors should be clear on what hypothesis is referred here and provide clear citations.

The hypothesis that we referred to is the "Neurodevelopmental Hypothesis" of schizophrenia. We revised this sentence for clarity as follows: “Gestational MAM administration is utilized to model prenatal neurodevelopmental disorders, providing an animal model that reflects the neurodevelopmental hypothesis of schizophrenia” (page 5, line 132-134).

For reproducibility, please address below issues:

7. For Figure 1, please include a representative figure showing the morphology of the recorded neurons, their laminar location, border of ACC, and location of stimulating electrode and recording pipette.

In the revised manuscript, we have added a new representative image to Figure 1b, d and o. The image shows both the Infrared DIC image interlaced with fluorescence of the recorded neurons and fluorescent renderings of the same neuron. Key elements, such as the laminar location, the border of the ACC, as well as the positions of the stimulating electrode and recording pipette, are clearly indicated.

8. For optogenetics, please clarify whether a laser or a LED was used as the light source. What was the light power? What was the filter setting?

We used a laser, with a specific wavelength (470nm) and a power setting of 10mW, as the light source for our optogenetic stimulation. We didn't employ any filter settings for the laser used in our study. We have now detailed these settings in the revised Methods (page 7, line 172-173; page 18, line 504).

9. After last revision, I appreciate that the authors now provide total animal and total cell number information for each figure, but on average how many cells are sampled from each animal? Was there effort to limit pseudo-replication? How many cells can come from a single animal? From the supplementary dataset, it appears that only 1-3 cells were from each animal, but this needs to be clearly stated for rigor and reproducibility.

Extending our apologies for any oversight in the previous versions, we wish to clarify that all raw data utilized in the current study, including the sample size pre group and cell number from each mouse, are included in the Supplementary Dataset file. For whole-cell recordings,

an average of 3 neuronal cells (ranging from 1 to 5) were sampled per animal, with the majority falling within the range of 2-4 neurons. This information is clearly stated now in the Methods (page 21, line 605-606). To limit pseudo-replication, we ensured that the cells came from at least 4 animals per group and were distinct from each other in each and every experiment.

There is still room for improvement for grammar and clarity, for example:

10. Line 367: “dose” should be “does”

11. Line 410: “couldn’t” should be “could not”

We have carefully overlooked the grammar and style in the revised manuscript and corrected the typos.

Reference:

- 1 Kuzmenkov, A. I. *et al.* Apamin structure and pharmacology revisited. *Frontiers in pharmacology* **13**, 977440, doi:10.3389/fphar.2022.977440 (2022).
- 2 Deignan, J. *et al.* SK2 and SK3 expression differentially affect firing frequency and precision in dopamine neurons. *Neuroscience* **217**, 67-76, doi:10.1016/j.neuroscience.2012.04.053 (2012).
- 3 Ferreira-Neto, H. C. & Stern, J. E. Functional coupling between NMDA receptors and SK channels in rat hypothalamic magnocellular neurons: altered mechanisms during heart failure. *J Physiol* **599**, 507-520, doi:10.1113/JP278910 (2021).
- 4 Behnisch, T. & Reymann, K. G. Inhibition of apamin-sensitive calcium dependent potassium channels facilitate the induction of long-term potentiation in the CA1 region of rat hippocampus in vitro. *Neurosci Lett* **253**, 91-94, doi:10.1016/s0304-3940(98)00612-0 (1998).
- 5 Johnston, M. V., Grzanna, R. & Coyle, J. T. Methylazoxymethanol treatment of fetal rats results in abnormally dense noradrenergic innervation of neocortex. *Science* **203**, 369-371, doi:10.1126/science.32620 (1979).
- 6 Chalkiadaki, K. *et al.* Development of the MAM model of schizophrenia in mice: Sex similarities and differences of hippocampal and prefrontal cortical function. *Neuropharmacology* **144**, 193-207, doi:10.1016/j.neuropharm.2018.10.026 (2019).
- 7 Sonnenschein, S. F. & Grace, A. A. Insights on current and novel antipsychotic mechanisms from the MAM model of schizophrenia. *Neuropharmacology* **163**, 107632, doi:10.1016/j.neuropharm.2019.05.009 (2020).

REVIEWERS' COMMENTS

Reviewer #4 (Remarks to the Author):

I am pleased to note that the authors have diligently addressed all of the concerns in my previous review. The newly added discussion strengthened the logical flow of the argument, and the revised citation enhanced the clarity and coherence of the research presented.

Clearly, the authors have demonstrated a commitment to enhancing the quality of the manuscript, and I have no further comments to add.